# Hot Spots for the Use of Intranasal Insulin: Cerebral Ischemia, Brain Injury, Diabetes Mellitus, Endocrine Disorders and Postoperative Delirium

**DOI:** 10.3390/ijms24043278

**Published:** 2023-02-07

**Authors:** Alexander O. Shpakov, Inna I. Zorina, Kira V. Derkach

**Affiliations:** Sechenov Institute of Evolutionary Physiology and Biochemistry, Russian Academy of Sciences, 194223 St. Petersburg, Russia

**Keywords:** intranasal insulin, brain, diabetes mellitus, cerebral ischemia, postoperative delirium, traumatic brain injury, gonadal axis, thyroid axis, neurodegeneration, neuron

## Abstract

A decrease in the activity of the insulin signaling system of the brain, due to both central insulin resistance and insulin deficiency, leads to neurodegeneration and impaired regulation of appetite, metabolism, endocrine functions. This is due to the neuroprotective properties of brain insulin and its leading role in maintaining glucose homeostasis in the brain, as well as in the regulation of the brain signaling network responsible for the functioning of the nervous, endocrine, and other systems. One of the approaches to restore the activity of the insulin system of the brain is the use of intranasally administered insulin (INI). Currently, INI is being considered as a promising drug to treat Alzheimer’s disease and mild cognitive impairment. The clinical application of INI is being developed for the treatment of other neurodegenerative diseases and improve cognitive abilities in stress, overwork, and depression. At the same time, much attention has recently been paid to the prospects of using INI for the treatment of cerebral ischemia, traumatic brain injuries, and postoperative delirium (after anesthesia), as well as diabetes mellitus and its complications, including dysfunctions in the gonadal and thyroid axes. This review is devoted to the prospects and current trends in the use of INI for the treatment of these diseases, which, although differing in etiology and pathogenesis, are characterized by impaired insulin signaling in the brain.

## 1. Introduction

After the discovery of insulin by Frederick Banting and Charles Best in 1921, it was found that this hormone is produced by pancreatic β-cells and, when administered to the organism, has a hypoglycemic effect, controlling glucose homeostasis. For a long time, it was believed that the functions of insulin were limited to the control of metabolism in the peripheral tissues, while the brain was not considered as its target. However, the discovery of insulin and its receptor in the brain by the group of Havrankova [1,2], and further study of the components of the insulin signaling system in neurons and glial cells of the brain, demonstrated its key role in the regulation of the central nervous system, the control of behavior and cognitive functions, as well as the central regulation of peripheral metabolism and the functions of the endocrine and other systems.

Following work on the role of insulin in the brain, evidence has been obtained that impaired insulin signaling in neurons and glial cells, both due to central insulin resistance (IR) and brain insulin deficiency, may be one of the main causes of neurodegenerative diseases and is closely associated with many metabolic and endocrine disorders, including type 2 diabetes mellitus (T2DM), metabolic syndrome (MS), and obesity. A detailed study of the role of brain insulin in the etiology and pathogenesis of the nervous and endocrine diseases led to the concept that the widespread neurodegenerative disease, Alzheimer’s disease (AD), is another form of diabetic pathology, referred to as type 3 diabetes. This form of diabetes is characterized by decreased insulin signaling in the CNS and cognitive deficit [3]. This concept was further confirmed and developed [4,5,6,7]. It has been established that impaired glucose homeostasis in the brain caused by central IR is one of the important causes of neuronal death and impaired synaptic plasticity in AD [8,9,10]. Close relationships between AD and T2DM, which is characterized by systemic IR, are shown. There are many works that the occurrence of AD in patients with T2DM and MS is significantly increased; as a result of which, these metabolic disorders are one of the risk factors for AD and other cognitive impairments [11,12,13,14,15].

Not surprisingly, as soon as approaches to restore and enhance insulin signaling in the brain began to be developed, they immediately start to be tested for the treatment of AD, mild cognitive impairment and other neurodegenerative diseases. The role of the “first violin” here is assigned to intranasally administered insulin (INI), the use of which allows you to quickly increase the level of the hormone in the brain and thereby stimulate insulin signaling in neurons [16,17,18,19]. Currently, the intranasal route of drug administration is being actively studied and considered for use in the clinic for the delivery of various hormones, growth factors, and pharmacological agents [20,21,22,23]. The pharmacokinetics and pharmacodynamics of INI have been extensively studied in the rodent models [16,17,18,19] and in human [24]. INI, at doses used in human clinical trials to treat patients with T2DM, AD, and mild cognitive impairment, does not significantly alter the blood levels of insulin or glucose, which indicates that there is a little risk of acute hypoglycemic episodes that can occur with subcutaneous and intravenous insulin injections, which may be dangerous [25,26,27,28,29].

To date, there is a large amount of experimental data, as well as clinical observations, indicating the effectiveness of INI in the treatment of AD, Parkinson’s disease, and mild cognitive impairment, as well as neuropathies [25,26,27,28,29,30,31,32,33,34,35,36,37,38,39,40,41]. It has also been shown that INI can improve memory in healthy subjects [42,43,44], which can be used to restore their cognitive abilities in case of overwork, psycho-emotional exhaustion, sleep restriction, and other physiological conditions leading to transient, reversible cognitive decline, and mood deterioration. Since numerous studies indicate the safety of INI, including its use at relatively high doses [28,29,34,45,46,47,48], it can be used not only in the treatment of overt pathology but also in the prevention of mild cognitive impairment to improve mood and to increase learning ability, resistance to stress, and prolonged physical activity [45,47,48,49]. These data indicate a pronounced neuroprotective effect of INI, its ability to increase the viability of neurons and glial cells, and restore the functional interaction between brain signaling systems, providing integration between different brain structures and improving the central regulation of physiological functions. They also indicate the ability of INI to improve metabolic processes in the brain, including glucose uptake and metabolism, which prevents the hypometabolic states that are characteristic of brain damage and neurodegenerative disorders.

Data on the effectiveness of INI in the treatment of AD and other cognitive dysfunctions open up prospects for its use in the treatment of other pathologies, which are also, at least in part, due to functional changes in insulin signaling in the brain. These pathologies include ischemia (stroke) and brain injury, in which there is hyperactivation of apoptotic and inflammatory processes in neurons and glial cells. This leads to severe, extensive neurodegenerative changes, impaired synaptic plasticity, and disintegration of brain signaling systems. INI can be effective in diabetic pathology not only for preventing and correcting cognitive disorders caused by diabetes mellitus (DM) but also for normalizing feeding behavior, energy homeostasis, and functions of the endocrine and other systems, the regulation of which directly or indirectly depends on the brain insulin system. Significant expectations are now associated with the use of INI for the correction and prevention of CNS dysfunction caused by anesthesia, the so-called postoperative delirium, which develops in the postoperative period and significantly worsens the prognosis of surgical interventions.

Despite great progress in the recent years, the mechanisms, targets, and efficacy of INI in the treatment and prevention of cerebral ischemia, brain injury, DM, DM-associated endocrine disorders, and postoperative delirium are still only poorly understood, and its therapeutic potential for the treatment of these pathologies remains greatly underestimated. Meanwhile, this line of INI research opens up new opportunities for understanding the functions of insulin and its system in the brain and also significantly expands the scope of INI clinical application. This review is devoted to the analysis and generalization of currently available literature data and our results on the possible use of INI for the treatment and prevention of cerebral ischemia and trauma brain injury, different types of DM, and disorders caused by anesthesia. This is preceded by a brief overview of the role and signaling mechanisms of action of insulin, including in the brain.

## 2. Insulin and Insulin Signaling System in the Brain and Intranasal Delivery of Insulin to the Brain

### 2.1. Sources of Insulin in the Brain

The origin of insulin in the brain is associated mainly with the intake of hormone produced by β-cells from the bloodstream due to its transport through the blood–brain barrier (BBB). However, a high level of insulin in the brain during the neonatal period of development, when the insulin-producing function of the pancreas is either absent or still insufficient for the synthesis of the required amounts of the hormone, indicates the possibility of insulin synthesis in the brain de novo, at least at the early stages of ontogeny. This is evidenced by the presence of mRNA for proinsulin in a number of brain regions in rat embryos and in newborn rats and rabbits [50], as well as in the culture of neurons obtained from various regions of the rabbit embryonic brain [51]. In this regard, it is important to note that neurons, such as pancreatic β-cells, are electrically excitable and respond by depolarization and exocytosis to the action of hormones and to an increase in glucose levels, which is very important for the synthesis and secretion of insulin by them, since these processes require depolarization of ATP-sensitive potassium channels functionally active in both cell types [52,53].

The entry of insulin circulating in the blood into the brain through the BBB is carried out using three main mechanisms. The first includes the transport of insulin to brain structures using receptor-mediated endocytosis (transcytosis), which is based on the binding of the hormone to insulin receptors (INSRs) located on the surface of endothelial cells, the subsequent internalization of insulin-receptor complexes included in vesicles into cytoplasm, and their transfer to the opposite side of the endothelial cell, followed by exocytosis on its abluminal side and the release of insulin towards the endings of neurons or glial cells [54,55]. The decisive role of INSR in this process is evidenced by the fact that in mice with a knockout of the *Insr* gene in endothelial cells, the permeability of the BBB for insulin deteriorates sharply, and as a result, insulin signaling in the hypothalamus, hippocampus, and prefrontal cortex is impaired [56].

There is competition between insulin and insulin-like growth factor-1 (IGF-1), another member of the insulin family, for receptors that mediate the transport of insulin across the BBB, since IGF-1 at higher than physiological concentrations is able to suppress insulin transcytosis [57]. Insulin transcytosis is regulated by hormones, lipids, and vasodilation factors. It significantly depends on the physiological parameters, weakens with obesity and MS, and increases with inflammatory diseases and insulin-deficient forms of DM, although the mechanisms in each case differ and are not always investigated [55,58,59,60,61,62,63,64,65]. Astrocytes and pericytes, which form close contacts with vascular endothelial cells, play an important role in modulating transendothelial insulin transport across the BBB [66]. The loss of pericytes or a violation of their structure leads to an increase in the permeability of the BBB and an impaired selectivity of transport of insulin and other hormones through the BBB [67,68,69].

Other mechanisms include passive insulin transport without INSR involvement through the fenestrated capillaries and ependymal cells of the median eminence, located below the mediobasal hypothalamus, ventral to the third ventricle and adjacent to the arcuate nucleus of the hypothalamus [70]. The transport of insulin across the median eminence is regulated by tanycytes, highly specialized ependymal cells that line the ventrolateral wall and the floor of the third ventricle and are involved in the exchange of substances between the blood and cerebrospinal fluid [71,72]. The presence of tight junctions between adjacent tanycytes provides a physical barrier that controls the transfer of insulin from portal capillaries and cerebrospinal fluid to the hypothalamus [73,74]. Remodeling of the hemato-hypothalamic barrier, for example, due to changes in blood glucose levels, affects the transfer of insulin and other circulating hormones through it to the arcuate nuclei of the hypothalamus [73,75]. In this regard, it should be noted that insulin enters the arcuate nuclei of the hypothalamus mainly through its passive transport with the participation of tanycytes, and not through receptor-mediated transcytosis [72].

### 2.2. Insulin Signaling System

A key component of the insulin signaling system in brain neurons and glial cells are INSRs, which are found in the hypothalamus, hippocampus, cerebellum, thalamus, cerebral cortex, olfactory bulb, dentate gyrus, motor cortex, and other regions [76,77,78,79,80,81,82,83,84,85]. In mammals, two INSR isoforms are present, the full-length (INSR-B) and the truncated (INSR-A). INSR-A is formed as a result of alternative splicing of the 11th exon by removing a 12 amino acid segment from the encoded sequence. These INSR isoforms are co-expressed in many tissues, but the ratio of INSR-A/INSR-B in the brain is much higher than at the periphery, which is why INSR-A is often referred to as the neuronal isoform of the receptor [86,87,88]. INSR-A and INSR-B are similar in structural organization and pharmacological characteristics, although they have a number of functional differences, including those due to differences in glycosylation [89,90,91]. Insulin is also able, albeit with low affinity, to bind to a hybrid receptor that includes the monomers of both INSR and IGF-1 receptor (IGF1R). The IGF1R is structurally and functionally similar to INSR, since the genes encoding these receptors, *Igf1r* and *Insr*, originate from a common ancestral gene and are part of an ancient, highly conserved signaling system in vertebrates and invertebrates involved in the regulation of cell metabolism, growth, and differentiation [92,93].

Both INSR isoforms are α_2_β_2_-heterotetramers in which extracellular α-subunits and transmembrane β-subunits are linked by disulfide bonds [94,95]. The α-subunit and the N-terminal segment of the β-subunit form the INSR ectodomain responsible for high-affinity insulin binding. The large cytoplasmic domain of the β-subunit contains a highly conserved tyrosine kinase domain that is stimulated after insulin binds to the ligand-binding site located in the α-subunit [94]. Binding of the α-subunit to insulin results in conformational rearrangements that affect the tyrosine kinase domain of the β-subunit and induce receptor autophosphorylation at the tyrosine residues, Tyr^1158^, Tyr^1162^ and Tyr^1163^, the main targets of the receptor tyrosine kinase. After autophosphorylation, the β-subunit interacts with regulatory and adapter proteins that contain phosphotyrosine-binding sites that can specifically interact with segments including phosphorylated Tyr^1158^, Tyr^1162^ and Tyr^1163^ (Figure 1). Depending on the pattern of this interaction, various signaling pathways are activated, resulting in different responses of the target cell to insulin [96].

After binding to the extracellular domain of INSR, insulin stimulates the intracellular receptor tyrosine kinase, which leads to the phosphorylation of three tyrosine residues (Y^1158^, Y^1162^, and Y^1163^), and further to the tyrosine phosphorylation of the IRS proteins and the adapter SHC protein. Through IRS proteins, the 3-phosphoinositide pathway is stimulated, including the enzyme PI3K catalyzing the synthesis of PI-3,4,5-P3 from PI-4,5-P2 and the effector serine/threonine protein kinase AKT. The targets of AKT-kinase are numerous enzymes and transcription factors that control cell growth, apoptosis, differentiation and metabolism, and AKT-kinase inhibits some of them (GSK3β, PGK1α, BAD, transcriptional factors of the FOX family), while others stimulate (PDE3b, PCK1, eNOS). AKT activates the protein kinase complex mTORC1, which leads to the stimulation of p70S6K kinase and a number of transcription factors involved in the regulation of protein and lipid synthesis. Along with this, AKT and PDK1 are also involved in the process of translocation of the insulin-dependent glucose transporter GLUT4 into the cell membrane, followed by GLUT4-mediated glucose uptake. Through tyrosine-phosphorylated IRS or SHC, insulin triggers a cascade of mitogen-activated protein kinases. The SHC protein in phosphorylated form is able to form a complex with GRB2 and SOS, which makes it possible to induce the exchange of guanine nucleotides in small G protein, p21Ras, converting it into an active GTP-bound form. Downstream components of the cascade of mitogen-activated protein kinases, ERK1/2 kinases, phosphorylate and thereby stimulate the activity of many transcription factors and enzymes, regulating the transcriptional activity of the genome. The functional activity of the insulin signaling cascade is under the control of a large number of negative regulators, the targets of which are INSR (PTP1B and TC-PTP), IRS proteins (PTP1B and JNK1), 3-phosphoinositides (PTEN, SHIP2), and AKT-kinase (PP2A and PHLPP1/2).

Insulin receptor substrates (IRS) proteins, which contain a large number of phosphotyrosine-binding sites, play a leading role in the transduction of the insulin signal from the receptor to intracellular effectors. The interaction of INSR with them activates several signaling pathways at once [96]. Despite a significant number of IRS protein isoforms, the IRS-1 and IRS-2 isoforms, which are ubiquitously expressed, are of the greatest importance for insulin signaling. At the periphery, IRS-1 is more involved in insulin signal transduction, while, in the CNS, the situation is different, and IRS-2 is most relevant for insulin signal transduction [97]. There is evidence that in the brain, the IRS4 proteins are involved in the transduction of signals generated by insulin and IGF-1. IRS4 can functionally replace IRS-1 and IRS-2, and in hypothalamic neurons, together with IRS-2, they are involved in the regulation of glucose homeostasis, feeding behavior, and energy expenditure [98,99]. In the N-terminal part of IRS proteins, the plekstrin-homology and phosphotyrosine-binding domains are localized, which ensure their interaction with INSR and association with the plasma membrane. The C-terminal part of IRS proteins contains a domain involved in specific interaction with various types of src-homology-2 (SH2)-domain-containing proteins responsible for the cell’s response to insulin [100]. Among the SH2-domain-containing proteins that are targets of insulin and IGF-1, the most important are phosphatidylinositol-3-kinase (PI3K), adapter protein GRB2, protein phosphotyrosine phosphatase SHP2, non-receptor tyrosine kinase Fyn, and suppressors of cytokine signaling.

After interacting with activated INSR, IRS proteins acquire the ability to bind to the phosphotyrosine-binding domain of the p85 regulatory subunit of PI3K and release the catalytic subunit of the enzyme, p110, which converts phosphatidylinositol-4,5-diphosphate to phosphatidylinositol-3,4,5-triphosphate (PI-3,4,5-P3), the most important second messenger [101] (Figure 1). PI-3,4,5-P3 induces the translocation of 3-phosphatidylinositol-dependent protein kinases of types 1 and 2 (PDK1/2), SIN1 protein, a component of the mammalian target of rapamycin complex 2 (mTORC2), and various isoforms of serine/threonine protein kinase AKT (AKT1, AKT2, or AKT3) to the plasma membrane to form an oligomeric complex, resulting in the activation of AKT [96]. In the CNS, the expression of AKT kinase isoforms is specific to both certain cell types and brain regions. AKT1 and AKT3 are mainly expressed in neurons, and, to the greatest extent, in the amygdala, hippocampus, and cerebral cortex, while AKT2 is predominantly expressed in astrocytes [102]. Membrane-bound PDK1 phosphorylates AKT at Thr^308^, while mTORC2 and DNA-dependent protein kinase (DNA-PK) at Ser^473^, and phosphorylation at two sites at once leads to complete activation of the enzyme [103]. The targets of AKT are phosphodiesterase 3B, phosphoenolpyruvate carboxykinase 1, and endothelial NO synthase, which are stimulated by AKT-induced phosphorylation, as well as peroxisome proliferator-activated receptor γ coactivator 1-α (PGC1α), glycogen synthase kinase-3β (GSK3β), and proapoptotic protein BAD, whose activity is reduced after AKT phosphorylation. AKT-induced phosphorylation of transcription factors of the forkhead box family (FOXO1, FOXK1) inhibits their activity and suppresses FOX-dependent gene transcription [96].

AKT kinase plays an important role in the control of cellular functions by regulating the expression of the genes responsible for cell growth, survival, differentiation, and metabolism. Along with this, it is involved in the translocation of the glucose transporter GLUT4, which provides insulin-dependent glucose uptake by cells into the plasma membrane [104]. AKT-induced phosphorylation of GSK3 at Ser^21^ (GSK3α) or Ser^9^ (GSK3β) results in GSK3 inactivation and blocks GSK3-mediated regulation of glycogen synthase activity, a key enzyme in glycogen synthesis that controls glucose homeostasis [105]. Along with the glycogen synthesis, GSK3 regulates the activity of a large number of transcription factors and their coactivators and corepressors, including NF-κB, Snail, Notch, BAD, and transcription factors of the FOX family. As a result, AKT-mediated inhibition of GSK3 makes a significant contribution to the effects of insulin on gene expression, apoptosis, autophagy, and cell differentiation [105,106,107,108]. In the CNS, GSK3β controls neuronal progenitor cell proliferation, neuronal polarity, and neuroplasticity [109,110]. GSK3β phosphorylates the tau protein, hyperphosphorylated forms of which are associated with neurodegenerative diseases, including AD. As a consequence, the weakening of insulin signaling in the brain, for example, in the conditions of neuron-specific knockout of INSR and IRS-2, promotes hyperphosphorylation of the tau protein and provokes neurodegeneration and cognitive deficit [111]. In addition, as noted above, transcription factors of the FOX family, which control various functions in the CNS, including energy homeostasis, feeding behavior, and locomotor activity, are the most important target of GSK3β [112,113]. The dysregulation of FOX-family factor activity by insulin, mediated through the 3-phosphoinositide pathway and GSK3β, leads to neurodegenerative diseases and metabolic disorders [114,115].

Another target of insulin is the mitogen-activated protein kinase (MAPK) cascade, which is stimulated by insulin through a signaling pathway including activated INSR, IRS proteins, GRB2 adapter protein, guanine nucleotide exchange factor SOS (Son of Sevenless), and p21ras, a small G protein (Figure 1). The GRB2/SOS complex stimulates the exchange of guanine nucleotides in p21ras, converting it into an active GTP-bound form, which triggers the MAPK cascade: Ras-protein→Raf-kinase→MEK1/2→ERK1/2. ERK1/2 activation and results in the increased activity of transcription factors and regulatory proteins, such as ribosomal protein S6, NF-κB, protein phosphatase-1 (PP-1), MYT1, Elk-1, cAMP-regulated transcription factor CREB, proto-oncogenes c-Fos and c-Jun, and transcription factors STAT family, which control a wide range of cellular processes [96,116]. Since the MAPK cascade is also regulated by other hormonal stimuli, at the level of various MAPKs, there is a crosstalk between insulin and other signaling systems, both in the brain and at the periphery.

AKT also phosphorylates TSC1/2 adapter proteins, and tuberous sclerosis proteins 1 (hamartin) and 2 (tuberin), which are negative regulators of mammalian target of rapamycin complex 1 (mTORC1). The result of this phosphorylation is the abolition of the blocking effect of TSC1/2 on the kinase activity of mTORC1 (Figure 1). Activated via AKT, mTORC1 controls the activity of factors that induce autophagy, as well as ribosomal p70-S6 kinase and transcription initiation factor eIF4E, which makes a significant contribution to the regulation of ribosome biogenesis, the processes of transcription, translation, and degradation of proteins [96,117]. mTORC1-dependent protein synthesis is extremely important for providing neuronal plasticity, as well as for the formation and maintenance of integrative connections between various brain structures [118]. In addition, mTORC1 activation enhances lipid synthesis through the activation of SREBP1 (Sterol regulatory element-binding transcription factor 1), which provides insulin control of lipid metabolism [119].

INSR activation, along with stimulation of intracellular signaling cascades, leads to the activation of negative feedback mechanisms that prevent hyperactivation of the insulin system and return it to its initial inactive state. Along with the stimulation of endocytosis of insulin-receptor complexes with their subsequent recycling or degradation in proteasomes, inactivation of the upstream components of the insulin signaling system is carried out by changing the phosphorylation of INSR and IRS proteins. A significant role in this is assigned to the dephosphorylation of tyrosine-phosphorylated forms of INSR and IRS proteins by tyrosine phosphatases, such as protein phosphotyrosine phosphatase 1B (PTP1B) and T-cell protein phosphotyrosine phosphatase (TC-PTP) [120,121,122]. Along with this, the activity of IRS proteins can be inhibited by their phosphorylation at serine and threonine residues with c-Jun N-terminal kinase-1 (JNK1) [123]. Thus, an increase in the expression and functional activity of phosphatases PTP1B and TC-PTP and JNK1 kinase, which reduces the efficiency of insulin signaling, is one of the causes of peripheral and central IR, including in obesity, MS, and T2DM [121,124]. In hyperphagia and obesity, hyperactivation of both tyrosine phosphatases is observed in the hypothalamus, which leads to a decrease in the sensitivity of hypothalamic neurons to insulin, and in prolonged consumption of saturated fatty acids, JNK1 activity increases in the hypothalamus and other regions of the brain, which reduces the activity of IRS proteins and leads to central IR [124,125]. A decrease in JNK1 activity prevents the negative effects of a long-term high-fat diet on insulin sensitivity in the brain, as well as on insulin-mediated metabolic regulation and cognitive functions [126].

Along with the suppression of the activity of INSR and IRS proteins, the negative regulation of insulin signaling can be carried out for its downstream effector components, as in the case of the 3-phosphoinositide pathway. Several phosphatases can act as negative regulators of this pathway, such as protein phosphatase PP2A, phosphatase PTEN (Phosphatase and TENsin homolog), and protein phosphatases PHLPP-1 and PHLPP-2 (PH domain and Leucine-rich repeat Protein Phosphatases 1/2) [104]. Phosphatase PTEN prevents the synthesis of 3-phosphoinositides by hydrolyzing PI-3,4,5-P3, while phosphatases PP2A and PHLPP1/2 dephosphorylate AKT kinase, inhibiting its activity. Another negative regulator of the 3-phosphoinositide pathway is type 2 SH2-containing inositol-5’-phosphatase (SHIP2), whose expression has been shown in various regions of the brain [127]. It has been shown that the level of PTEN in the substantia nigra of patients with Parkinson’s disease is significantly increased, which is associated with enhanced neurodegeneration [128], and an increase in the content of PTEN increases brain damage after ischemia/reperfusion [129]. Hyperactivation of phosphatase PP2A in the brain, causing central IR and an imbalance in the regulatory effects of insulin and other neurohormones, is closely associated with neurodegenerative diseases, including AD and Parkinson’s disease [130].

It should be noted that all the main blocks of the brain insulin signaling system are involved in the implementation of the neuroprotective, neurotrophic, and neuromodulatory effects of INI and also mediate its effects on other hormonal and neurotransmitter systems that are integrated and coordinated with the brain insulin system.

### 2.3. The Intranasal Route of Insulin Delivery

Over the past decades, the intranasal route of administration has become relevant for the delivery of drugs to the CNS, bypassing the BBB. More than 30 years ago, William H. Frey II was the first to propose the use of intranasal administration of various biologically active substances, including polypeptide hormones and growth factors, for the treatment of Alzheimer’s disease and other diseases of the nervous system, and insulin was of the greatest interest among the studied hormones [131,132,133]. To date, there are already more than 50 clinical trials of INI in both healthy subjects and patients with CNS diseases and diabetes (https://clinicaltrials.gov/, accessed on 27 December 2022).

The main advantage of intranasal administration of drugs is the targeted delivery to the CNS of substances that are unable to penetrate the BBB due to their size, charge, or other physical, chemical and biological properties. This approach provides a rapid achievement of a therapeutic effect, as well as non-invasiveness and ease of use compared to oral or injection methods of drug delivery. An important feature of intranasal administration is the absence of passage through the hepatic barrier, which allows the use of lower doses of drugs that reduce side effects [16,20,21].

Significant progress in the study of the INI mechanisms involved in bypassing the BBB, its pharmacokinetics and pharmacodynamics, as well as the targets of INI in the brain, has been achieved through the use of animal models [16,17,18,19,134]. After nebulization in the nasal cavity, insulin enters the nasal mucosa, where it is transported to the brain by an intracellular route using receptor-mediated endocytosis via axons of olfactory and trigeminal nerves, which innervate the nasal cavities, as well as via extracellular pathway by interstitial fluid, but with different speeds. Retrograde axonal transport is rather slower [135] than an extracellular one, so it takes hours to reach the inner brain structures, and more likely that INI is primarily transported along extra-neuronal routes [16,136]. The extra-neuronal pathway rapidly allows insulin and its homolog IGF-1 to spread along the olfactory and trigeminal nerves and distribute within brain in the rostral to caudal direction [18,19,137]. Once in the CNS, insulin binds to brain INSRs and causes their activation, which is demonstrated by an increase in the level of tyrosine receptor phosphorylation [19]. This leads to activation of intracellular signaling pathways, including the PI-3-K/Akt signaling pathway, as mentioned above [17,18]. After 15–30 min after intranasal administration, insulin accumulation was shown in the olfactory bulbs, striatum, substantia nigra, brainstem, cerebellum, and, to a lesser extent, in the hippocampus and cerebral cortex [17,18]. According to data from other authors, large amounts of intranasally administered insulin were found in the cortex, cerebellum, hippocampus, and hypothalamus [19]. Thereafter, the amount of insulin in the brain gradually decreases but remains elevated even 6 h after its administration [18]. An increase in insulin levels after intranasal administration was also found in the cerebrospinal fluid and in a small amount in the serum [134,138,139]. With long-term (9 days) administration to mice, INI increased the level of glucose, adenosine triphosphate, and phosphocreatine in the brain of animals, which indicates its positive effect on energy metabolism in the brain [134]. In this regard, it should be noted that in humans, a positive effect of INI on improving glucose homeostasis and energy metabolism in the brain has been demonstrated, which is important for correcting hypometabolic conditions in neurodegenerative and other diseases [140].

In human studies, INI bypassed the BBB and detected in the cerebrospinal fluid at 10 min with peak at 30 min and remained significantly elevated at 80 min [24]. The serum insulin levels peaked in 10–20 min after its intranasal administration, then remained slightly elevated for approximately 1 h and returned to mean values 1.5–2 h after nebulization with an insignificant risk of hypoglycemia mainly in a fasting state [25,141]. The fast and unhindered entry of INI into the CNS is a great advantage of the intranasal route of hormone administration compared to other routes of its administration, which is important in pathologies that require emergency treatment, such as stroke and brain injury, as well as during pre- and postoperative procedures.

The data obtained from the use of INI in humans does not raise any significant concerns in its short-term and long-term use. On the contrary, most studies point to its positive effect on cognitive functions, good tolerability, and the absence of serious side effects that could become an obstacle to INI therapy. However, it should be noted that the chronic, long-term use of INI in a small cohort of patients still revealed some undesirable effects, which are mainly due to the procedure of spraying INI in the nasal cavity (rhinitis, mild epistaxis, sneezing, headaches, and predisposition to respiratory tract infections) [26,27,28,33,46]. It is important that these undesirable effects can be eliminated by optimizing both the procedure for intranasal nebulization of the hormone and the composition of the mixture for nebulization, since they are not caused by the toxic effect of INI on the nasal cavity and brain structures.

## 3. Intranasal Insulin and Brain Ischemia

Cerebral ischemia occurs as a result of various critical events and diseases, including surgery, resuscitation, traumatic brain injury, asphyxia, hemorrhages, embolisms, myocardial infarction, acute blood loss, and hypotension [142,143,144]. However, the main cause of damage to brain neurons is a deficiency of oxygen and nutrients in the CNS. The consequences of cerebral ischemia are determined by the level of damage and death of neurons and glial cells and depend on the degree of deterioration of blood flow in the brain tissue, the duration of the ischemic episode, and the area of the ischemic lesion. The metabolic and hormonal status of the brain and organism before cerebral ischemia is also of great importance for the severity of CNS damage caused by it [143]. Developing glutamate excitotoxicity, oxidative stress, mitochondrial dysfunction, inhibition of protein synthesis, and activation of proteolytic enzymes lead to the triggering of endoplasmic reticulum stress, cell death, and neuroinflammation. Prolonged brain ischemia leads to disintegration and irreversible changes in the brain signaling cascades responsible for the survival and functional activity of neurons, resulting in their death through apoptosis, necrosis, or autophagy [145]. Not only neurons but also glial cells are damaged during ischemia, which negatively affects the neuroprotective potential of glia. At the brain level, cerebral ischemia is accompanied by cerebral edema, damage to the integrity and permeability of the BBB, hyperactivation of microglia, and increased pro-inflammatory cascades [146,147,148,149]. During ischemia, compensatory mechanisms are activated that maintain the viability of neurons and reduce the intensity of pro-inflammatory and apoptotic processes in the CNS. These mechanisms include signaling cascades triggered by a large number of neurohormones, growth factors and cytokines with neuroprotective properties, and the insulin and IGF-1 signaling cascades are of key importance among them [145]. As a result, the use of hormonal regulators with neuroprotective properties, including insulin, is able to compensate for ischemic brain damage, which makes them promising drugs for the treatment of ischemia.

Despite many years of research in the field of treatment of ischemic stroke, there are currently no anti-ischemic neuroprotective drugs that are used clinically [150,151]. Since the pathophysiology of stroke is multi-faceted, it is promising to use neuroprotectors with a pleiotropic effect, which makes it possible to prevent several pathological processes in the brain at once. This predetermines the main direction of the search for pharmacological agents to treat postischemic dysfunctions of the nervous and other systems.

Hyperglycemia leads to more severe consequences [152,153,154], making glycemic control one of the most important steps in the treatment and rehabilitation of patients with cerebral ischemia [155]. In this regard, it should be noted that both hyperglycemia and hypoglycemia, including those induced by insulin therapy, are associated with a poor prognosis of ischemia [156,157]. In this regard, it is not surprising that patients with DM or pre-diabetes have an increased risk of ischemic and hemorrhagic stroke [158,159,160,161]. Along with this, about half of the patients who had an ischemic stroke later had impaired glucose homeostasis and IR [162,163]. T2DM increases the risk of recurrent stroke after the first episode of ischemia or stroke, so secondary prevention of stroke in patients with metabolic disorders is also an important problem [150,151]. There is evidence that weight loss, restoration of tissue sensitivity to insulin, and normalization of glucose homeostasis are very effective in stroke prevention [150,151,158,159,160,161].

All of the above indicates that injectable forms of insulin, which normalizes glucose levels in hyperglycemia, can be indicated for the correction of hyperglycemic conditions in diabetic patients with stroke, as well as in non-diabetic patients with impaired glucose homeostasis that developed as a result of a stroke. However, in the recent years, the focus of the use of insulin has shifted from its metabolic effects to neuroprotective effect and the ability of insulin to influence the functional activity of brain structures that have undergone ischemic damage. In connection with the foregoing, let us consider the currently available data on the effect of insulin and its functional homolog, IGF-1, on postischemic damage in animals with experimental ischemic and hemorrhagic stroke and hypoxia-ischemia.

The neuroprotective properties of insulin in ischemic injury have been studied for more than 30 years [164,165,166,167]. It has been shown that insulin with various delivery methods (subcutaneous, intraperitoneal, intravenous, and intracerebroventricular) positively affects the survival and cognitive functions of animals that have undergone cerebral ischemia [165,168,169,170,171,172,173,174,175,176]. The antiapoptotic signaling pathway, including IR/IRS/PI3K/AKT, plays a leading role in the implementation of the neuroprotective action of insulin. By activating AKT, insulin suppresses the release of cytochrome *C* from mitochondria, thereby preventing the translocation of the proapoptotic protein BAX to them [177,178], and also stimulates synthesis of neuroprotective proteins, including the antioxidant enzymes [179,180], which increases the survival of neurons and prevents cognitive dysfunctions in animals with cerebral ischemia [178]. The contribution of insulin to the improvement of cognitive functions during oxidative stress or excitotoxicity may also be due to its neuromodulatory effect, by changing the surface density and activity of the glutamate and γ-aminobutyric acid ionotropic receptors in neurons, as well as by regulating the membrane transport of glutamate and γ-aminobutyric acid, which allows optimizing their ratio in the extracellular space [180].

The vasoactive effect of insulin is widely known, which, along with its neuroprotective effect, plays an important role in the implementation of the protective functions of insulin in cerebral ischemia. AKT-dependent activation of endothelial NO-synthase under the action of insulin on the endothelium of cerebral vessels leads to changes in the vascular lumen and improves the cerebral blood flow (CBF) [181,182]. At the same time, depending on the dose of the hormone and the functional state of the CNS, insulin is able to both increase and decrease the lumen of the vessels, exerting a modulating effect on CBF [183,184]. In healthy subjects, an increase in insulin levels in the brain leads to increased CBF and improved blood supply to brain tissue [185].

Despite a significant number of studies on the effects of peripheral and intracerebroventricular insulin administration to correct the consequences of cerebral ischemia, there are relatively few data on the use of INI, despite the promise of this method for correcting ischemic damage [186,187,188,189,190,191,192]. It should be noted that intranasally administered IGF-1 has been used for the correction of damage caused by cerebral ischemia–reperfusion for 20 years [186,193,194,195,196,197,198,199]. The targets of action of insulin and IGF-1 in the CNS largely coincide, which is due to both the similar architecture of their signaling cascades and the ability of insulin and IGF-1 to activate the same receptors, including hybrid di/oligomeric forms of INSR/IGF1R [95]. In addition, both hormones have a similar pattern of regulatory effects on neurons and glial cells, exerting antiapoptotic and anti-inflammatory effects, as demonstrated for insulin in the use of INI for the treatment of neurodegenerative diseases, including AD [27,200,201]. All this allowed the group of Vera Novak to put forward a hypothesis about the prospects of using INI to correct cerebral ischemia [186]. It is important that INI has clear advantages over intranasally administered IGF-1, because, unlike insulin, IGF-1 is characterized by a very powerful mitogenic effect, which has significant risks of cell malignancy, and the IGF-1 dosage form is expensive and has very limited clinical experience. It is also important that INI not only has a neuroprotective effect on neurons and glial cells but also improves the brain blood supply, which is especially valuable in the conditions of cerebral ischemia [185]. The potential mechanisms of neuroprotective action of INI in cerebral ischemia are shown in Figure 2.

In 2022, Yuan Zhu and coauthors investigated the neuroprotective properties of INI and optimized the doses of the drug used to correct ischemic damage in hemorrhagic stroke in mice [192]. They showed that the most effective dose was 1.0 IU/mouse, while at a dose of 2.0 IU/mouse the neuroprotective effect of INI was much less pronounced. The authors attribute this to the dose-dependence of the regulatory effects of insulin on CBF, which, at high doses, can cause excessive vasodilation in the area of hemorrhage and, thereby, increase brain tissue damage. They indicate that the neuroprotective effect of INI in hemorrhagic stroke is mainly due to the activation of the 3-phosphoinositide pathway, including stimulation of AKT and inhibition of downstream GSK3β [192]. An earlier study examined the effect of INI on the metabolic processes in the brain of mice after subarachnoid hemorrhage [191]. INI treatment (1.5 IU/mouse; immediately after and 24 h and 48 h after exposure) increased the expression of glucose transporters GLUT-1 and GLUT-3 in the cerebral cortex, increased the brain glucose levels, and decreased the lactate/pyruvate ratio in the interstitial fluid, which indicates a partial normalization of glucose homeostasis in the brain. There was also a decrease in neuroinflammation and improvement in cognitive functions. According to the authors, INI has a neuroprotective effect by reducing metabolic distress and maintaining metabolic homeostasis [191].

The pathogenetic mechanisms of stroke and TBI are quite similar, and therefore the neuroprotective effects of INI observed in cerebral ischemia can also be realized in TBI. After spraying in the nasal cavity, insulin enters various regions of the brain, where it exerts its regulatory effects through binding to INSR and, possibly, to the hybrid receptor INSR/IGF1R, which are found in neurons, glial cells, and endotheliocytes. Insulin binding to these receptors leads to their tyrosine phosphorylation and activation, resulting in transphosphorylation reactions leading to tyrosine phosphorylation of IRS proteins and activation of SH2-domain-containing effector proteins interacting with IRS proteins. One of the main targets of IRS proteins is the PI3K, which triggers the 3-phosphoinositide cascade and through it activates the protein kinase AKT. The result of this is a change in the functional activity of a large number of cell effector systems and gene expression. Thus, insulin stimulates the synthesis of proteins and lipids, controls the processes of neurogenesis, axonogenesis, cell differentiation, autophagy, and apoptosis and regulates the mitochondrial function and biogenesis, stimulating mTOR activity. Normalization of mitochondrial function ensures a decrease in lipid peroxidation (LPO), which, in turn, leads to the restoration of functional activity of the membrane transporters, including Na^+^/K^+^-ATPase. The insulin-activated MAPK/ERK cascade is involved in the control of cell survival and proliferation, as well as in protein biosynthesis. Insulin increases the survival of neurons and glial cells by suppressing the activity of proapoptotic caspases and inhibiting mitochondrial apoptosis. Insulin-induced stimulation of NO-synthase (NOS) activity in neurons, astrocytes, and endotheliocytes leads to an increase in NO production and has a vasodilating effect on cerebral vessels, improving CBF. Insulin is able to increase the extracellular γ-aminobutyric acid (GABA) levels and the expression of ionotropic GABA_A_ receptors, and also stimulates the uptake of glutamate by astrocytes, which leads to inhibition of neuronal activity, including under the conditions of glutamate excitotoxicity. One of the results of the insulin action on brain cells is an increase in glucose uptake due to increased expression and improved translocation into the plasma membrane of the glucose transporters. Insulin promotes the accumulation of glycogen in astrocytes, which is the basis for maintaining glucose homeostasis in the CNS. Intranasal (as well as intracerebral) administration of insulin reduces the production and release of pro-inflammatory cytokines and the formation of radical oxygen species (ROS) by microglia, and stimulates the expression of anti-inflammatory cytokines and the neurotrophic factor BDNF in astrocytes. Since functionally active IGF1R and INSR/IGF1R receptors are present in neurons and glial cells, intranasal administration of IGF-1, structural and functional homolog of insulin, can also have a protective effect on them, including suppression of mitochondrial apoptosis, neuroinflammation, and ROS overproduction. All these mechanisms, although to varying degrees, are involved in the neuroprotective and anti-inflammatory effects of INI and intranasally administered IGF-1, providing their restorative effect on cognitive, vestibulomotor, and somatosensory functions impaired in cerebral ischemia and TBI. However, it should be noted that, in contrast to neurons, data on the protective effects of insulin and IGF-1 in microglia and oligodendrocytes during ischemia and TBI remain poorly understood.

In 2018–2022, we carried out a set of studies on the ability of INI to influence metabolic processes and oxidative stress during bilateral global forebrain ischemia-reperfusion in rats [187,188,189,190]. At doses of 0.25 and 0.5 IU/rat, INI showed antioxidant and neuroprotective effects on the brain of rats with ischemia, and also prevented oxidative inactivation of Na^+^/K^+^-ATPase and accumulation of lipid peroxidation (LPO) products in the brain. The antioxidant effect of INI was carried out at the level of regulation of the expression of genes encoding antioxidant enzymes, as indicated by an increase in the gene expression for catalase and types 1 and 2 superoxide dismutases in the cerebral cortex of ischemic rats [187]. INI-induced normalization of LPO products content can be mediated by both an increase in the expression of antioxidant enzymes and an increase in the glutathione level [188]. This is supported by the data of in vitro experiments on an increase in the level of total glutathione during the insulin action on the primary culture of cerebral cortex neurons under the conditions of oxidative stress, which was due to an increase in the activity of glutathione reductase and a decrease in the activity of glutathione peroxidase [202].

One of the mechanisms to counter cerebral ischemia and hypoxia is the activation of the hypoxia-induced factor-1α (HIF-1α) and HIF-1α-dependent genes encoding proteins that promote the adaptation of brain cells to hypoxia. Thus, upon activation of HIF-1α, there is an increase in the expression and activation of proteins that ensure cell proliferation and survival, as well as an increase in iron metabolism and the synthesis of glycolytic enzymes and other metabolic regulators [203,204]. Since AKT kinase is the most important component of the signaling cascade that activates HIF-1α, the increase in the level of insulin and IGF-1 and the activation of their signaling cascades in the brain are considered as one of the main mechanisms for the activation of HIF-1α-dependent pathways during ischemia–reperfusion [205]. Thus, there is every reason to believe that INI, through the IR/IRS/PI3K/AKT signaling pathway, can stimulate HIF-1α activity, facilitating the adaptation of neurons and glial cells to cerebral ischemia, although this requires further research.

According to our data, INI demonstrates a neuroprotective effect during ischemia at a relatively low dose of 0.25 IU/rat [187,189,190], which is 4 times lower than the effective dose of INI, described in the work of Yuan Zhu and coauthors for mice (1.0 IU) [192]. Taking into account the peculiarities of pharmacological regulation in rats and mice, these doses can be considered as equivalent. On the other hand, a 2-fold increase in the dose (up to 0.5 IU/rat) in our case did not lead to a decrease in the neuroprotective effect of INI, as was shown for mice [187,192]. It should be noted that with other delivery methods, such as intravenous and subcutaneous, the neuroprotective effect of insulin was manifested at significantly higher doses (4–12 IU/rat) [182]. The effective doses of INI used by us are comparable to those for intracerebroventricular administration of the hormone. In this regard, it is noteworthy that INI at doses of 0.25–0.5 IU/rat clearly restored the metabolic and hormonal parameters, as well as cognitive functions, in rats with experimental models of DM [206,207,208,209] and MS [210]. Our studies on the effects of INI in DM and MS open up prospects for stroke prevention in patients with these diseases, as well as in patients who have undergone cerebral ischemia.

Another direction of research on the effectiveness of INI and intranasally administered IGF-1 in experimental ischemia is the study of the potentiation of its neuroprotective effect when used together with other drugs that have neuroprotective properties. In rats with ischemia, we have shown an increase in the antioxidant effect of INI when it is co-administered with α-tocopherol, a bioactive component of vitamin E, which may indicate a synergism in the neuroprotective action of insulin and α-tocopherol on neurons [189]. These data were supported by our results of in vitro experiments, which showed an increase in the neuroprotective effect of insulin in the presence of α-tocopherol in the primary culture of rat cerebral cortex neurons under the conditions of oxidative stress induced by hydrogen peroxide [189]. Other authors have shown that intranasally administered cytokine erythropoietin in ischemic stroke in mice has a powerful neuroprotective effect on brain structures when it is combined with IGF-1. This is manifested in a significant reduction in brain infarct volume and improvement in neurological function within 90 days after occlusion of the middle cerebral artery [196]. The study of the distribution of [^125^I]-IGF-1 in the ischemic brain of mice after intranasal administration showed that this growth factor is found in the ischemic zone of the brain 20 min after administration and subsequently accumulates in it, and upon intranasal administration, the concentration of IGF-1 in the brain is as in both healthy and ischemic mice was higher than with intravenous, subcutaneous, and intraperitoneal routes of IGF-1 administration [196]. These data can be translated with high reliability into INI, given the structural similarity of the peptides of the insulin group, and indicate a high bioavailability of various brain regions during intranasal administration of insulin and IGF-1, including in ischemia.

During cerebral ischemia, the white matter of the brain is damaged, which is associated with severe neurological consequences, especially in newborns [211]. The main target of damaging effects during acute ischemia and hypoxia are immature oligodendrocytes, which are involved in the myelination of nerve fibers. In the perinatal period, damage to the neural pathways and precursors of oligodendrocytes leads to periventricular leukomalacia and long-term demyelination [212]. In the mature brain, there are populations of glial cells (adult progenitor cells) that retain mitotic activity and serve as a source for myelin-forming oligodendrocytes. Their damage during ischemia leads to a long-term loss of myelin, followed by dysfunction of the white matter of the brain [213]. Unfortunately, there are currently no data on the effect of INI on the white matter of the brain in ischemic injury. At the same time, a number of studies have shown the effectiveness of the protective effects of intranasally administered IGF-1 on the white matter of the brain under the conditions of hypoxia and ischemia [197,214]. During hypoxia and neuroinflammation induced by intracerebroventricular administration of lipopolysaccharide to five-day-old rat pups, intranasal administration of IGF-1 one and two hours after exposure significantly reduced damage to the white matter and had antiapoptotic and anti-inflammatory effects, reducing the intensity of neurodegenerative processes and improving cognitive functions [197]. Intranasal administration of IGF-1 to rat pups subjected to postnatal hypoxia-ischemia significantly increased the survival of immature oligodendrocytes by suppressing the activity of proapoptotic caspase pathways and improved their myelination and proliferative activity. This led to the restoration of neurological functions impaired in young rat pups due to ischemic exposure [214].

Despite the fact that the population of immature oligodendrocytes in the white matter of the adult human brain is relatively small, nevertheless, the existing cell pool can contribute to remyelination after injury or ischemia. During ontogenetic development of the brain, immature oligodendrocytes migrate from the ventricular zone to their destination, where they differentiate with the formation of myelin sheaths. However, after a traumatic brain injury, they can migrate to the damaged area, where they contribute to the restoration of myelin [215]. Along with this, an intensification of proliferation of oligodendrocytes after cerebral ischemia has been shown, although the mechanisms of this remain poorly understood [216,217,218]. Since IGF-1 and insulin have, although to varying degrees, the ability to stimulate cell proliferation and migration, when administered intranasally, they can be very effective in restoring the pool of oligodendrocytes and their migration to areas of the brain, damaged as a result of ischemic exposure. Since the survival and growth of oligodendrocytes and the maintenance of the myelin sheath in the infarcted area largely depend on the activation of the transcription factor CREB, which is one of the targets of insulin [219,220], INI-mediated stimulation of CREB in oligodendrocytes can lead to the restoration of the functions of the white matter, impaired during cerebral ischemia.

The central IR characteristic of AD is associated with degeneration of the white matter of the brain, which is due to the loss of myelin and myelinated fibers, dysfunction or death of oligodendrocytes, hyperactivation of astrocytes, severe gliosis and microvascular dysfunction, and this indicates an important role of brain insulin in functioning of oligodendrocytes [221]. Accordingly, preventing the death of oligodendrocytes and stimulating their growth, as well as normalizing the activity of astrocytes and microglia, can play an important role among the reasons for the high effectiveness of INI in the treatment of patients with AD [39,222]. With regard to IGF-1, there is evidence that IGF-1 has a protective effect on the functioning of astrocytes, since a decrease in the level of IGF-1 or a knockout of the gene encoding IGF1R in astrocytes leads to a decrease in the rate of uptake of glutamate by astrocytes and provokes glutamate excitotoxicity [223]. It has also been established that disruption of IGF-1 signaling pathways in astrocytes leads to changes in glucose homeostasis, resulting in the impaired synaptic transmission and metabolic activity in neurons that depend on glutamine and lactate, which come from astrocytes [224]. Glucose homeostasis in astrocytes is also regulated through INSR [85], since knockout of these receptors in hypothalamic astrocytes leads to impaired glucose metabolism and changes in food intake [225,226]. At the same time, the effect of insulin on glucose homeostasis in astrocytes is carried out in close interaction with IGF-1 [227]. In vitro experiments have shown that shutdown of INSR in astrocytes leads to compensatory activation of IGF1R [228]. This may indicate that intranasally administered insulin and IGF-1, both in the correction of is chemic damage and other brain dysfunctions, can largely replace each other.

The Stroke Treatment Academic Industry Roundtable (STAIR) Association has published recommendations to be followed when searching for and validating neuroprotective drugs to treat stroke [229]. They include the study of the dose-dependent effect of drugs and the duration of the therapeutic window, provide for the need to test their effectiveness in various models of ischemia and ischemia–reperfusion using several animal species (rodents, cats, primates, etc.), and also include an assessment of the influence of age, gender and other factors. When searching for and developing neuroprotectors, it should be taken into account that the mechanisms of damage to white matter and neurons may differ, and it is also necessary to assess the degree of damage to the nervous tissue of the young, adult, and aging brain during hypoxia-ischemia and its ability to recover [230,231,232,233]. According to these recommendations, the current studies of INI and intranasally administered IGF-1 in cerebral ischemia are not without drawbacks and limitations. Most of them were carried out using young animals, and without taking into account the predisposition to stroke in the case of cardiovascular and metabolic pathology [234]. There are no dose-response studies of the INI effect and its therapeutic window. It should also be taken into account that the results obtained in animals cannot be fully translated to humans. In animal studies, the time of onset of cerebral ischemia, its duration and volume of cerebral infarction, as well as the presence of reperfusion and its duration, are accurately determined, while in the clinic the possibility of accurately assessing these indicators is not high. Along with this, in connection with the use of thrombolysis in stroke treatment protocols, there is a need to investigate the potential interaction between neuroprotectors and thrombolytics, which in the case of INI has not yet been studied.

Gender studies are important for the choice and optimization of a neuroprotective strategy, including for INI [235]. Animal studies have shown that the volume of cerebral infarction in females who have undergone arterial occlusion is significantly lower than in males [236]. Apparently, estrogens and progesterone significantly affect the severity of brain damage, since estrogen preparations reduce the severity of the consequences of ischemia in both males and females [237]. However, studies on the effect of gender on the protective effects of INI in cerebral ischemia are currently lacking.

## 4. Intranasal Insulin and Brain Injury

Traumatic brain injury (TBI) in most cases leads to long-term cognitive deficit, which is currently poorly or not treatable at all [238]. Even mild TBIs in 15% of patients lead to cognitive deficits in the long term [239]. TBI is based on a three-phase response of the brain to an external damaging effect [240]. The first phase corresponds to the development immediately after injury of a hypermetabolic state caused by a powerful release of ions and glutamate, which is due to a sharply increased demand for energy. The second phase corresponds to a significant decrease in the absorption of glucose by the brain tissue, which leads to an energy deficit in neurons and is the main cause of brain damage and triggering neurodegenerative processes. The long-term consequences of TBI, including the subsequently developing cognitive deficit, largely depend on the duration and severity of this phase. The final phase includes a gradual return of glucose uptake by the brain to normal values [240]. It is important that the restoration of metabolism can occur only in some regions of the brain, while other regions are characterized by a decrease in metabolic status over a long period, years after TBI, even in patients with a mild form of this injury [241,242]. The disintegration of functional connections between various brain structures, the decrease in neuronal plasticity in them, and the activation of neuro-inflammatory processes lead to dysfunctions of the CNS and an imbalance of metabolic processes at the periphery [243,244,245].

In animals with experimental TBI, a decrease in brain sensitivity to insulin was found, which may be a consequence of glutamate excitotoxicity induced by traumatic exposure and activation of pro-inflammatory processes in the brain [244,246,247]. With a single traumatic impact, a decrease in the brain sensitivity to insulin is observed within 7 days, and with repeated trauma, up to 28 days [246]. At the same time, in obese mice, which initially had impaired insulin signaling in the brain, the consequences of TBI were expressed to a much greater extent. They were accompanied by a significant weakening of the stimulatory effect of insulin on the INSR/IRS/PI3K/AKT signaling pathway, pronounced neuroinflammation, a significant deterioration in learning ability, a sharp decrease in memory, and a severe anxiety-depressive state [246].

A study of glucose uptake in the brain of rats showed that after a mild TBI, its changes begin to be registered as early as 3–6 h after the traumatic impact and persist to some extent for 5–10 days, and this effect was specific both for certain regions of the brain and for certain types of neurons and glial cells [248,249,250]. The development of post-traumatic complications is largely due to insulin sensitivity of the brain, which decreases with obesity and a number of other disorders, as well as the effectiveness of integrative interactions between brain regions, which can significantly affect insulin sensitivity and glucose uptake by neurons and glial cells [244,250]. Since after a single TBI, there is a transient weakening of the sensitivity of certain regions of the brain to insulin, it seems quite logical that repeated TBIs aggravate the central IR, especially when they occur in the time window of a decrease in insulin sensitivity caused by previous TBI. In 2022, a model of lateral fluid-percussion injury in rats showed that 4 months after traumatic exposure, glucose uptake was reduced in the hippocampus, ipsilateral striatum, and frontoparietal cortex [251]. Decreased glucose uptake in the stratum was positively correlated with deterioration in spatial memory [251].

All of the above points to the potential ability of INI to improve glucose homeostasis in the brain and compensate, at least in part, for the hypometabolic state that develops after TBI [244]. This is supported by the data on INI effect on adult male Sprague Dawley rats with moderate controlled cortical injury. Treatment of animals with INI (Humulin R-100, 6 IU/rat) was started 4 h after traumatic exposure, and then INI treatment at the same dose was continued for two weeks [252]. As a result, glucose utilization was normalized in the hippocampus of rats treated with INI, which was accompanied by a decrease in the hippocampal lesion area, a decrease in neuroinflammation mediated by microglia, and an improvement in memory and spatial orientation [252].

The prospects for the use of INI in TBI are also indirectly evidenced by the fact that moderate and severe TBI is significantly associated with AD, which is one of the targets of INI [253]. The severity of brain damage is positively correlated with the incidence of AD. It should be noted that a close relationship is also observed between AD and Parkinson’s disease, which have many common molecular causes, including metabolic changes in the brain that provoke neurodegeneration, and both are therapeutic targets for INI [254]. It should also be noted that the pathogenetic mechanisms of stroke and TBI are quite similar, and therefore the previously described neuroprotective effects of INI for cerebral ischemia can also be manifested in TBI [255]. The possible mechanisms and targets of neuroprotective action of INI in TBI are summarized in Figure 2. Thus, the general pathogenetic factors of CNS diseases may a priori suggest the potential effectiveness of INI for the treatment of TBI and the prevention of its negative consequences.

## 5. Intranasal Insulin and Diabetes Mellitus

DM is characterized by significant changes in the functional activity of the insulin signaling system, and this is due to both absolute or relative insulin deficiency in type 1 DM (T1DM) and the development of IR in T2DM. In both cases, the result is a weakening of insulin signaling pathways both in the brain and at the periphery. To restore them, pharmacological approaches can be used to compensate for insulin deficiency (insulin replacement therapy) or to reduce IR (metformin and other drugs that increase insulin sensitivity). Since insulin in the brain is involved in the regulation of the functions of neurons and glial cells, controls the integrative relationships between different brain regions, and mediates the central regulation of physiological and metabolic processes at the periphery, an impaired brain insulin signaling in DM leads to a wide range of disorders. As a result, the use of INI, which activates the insulin system in the brain, may be useful for preventing cognitive deficits in patients with DM, as well as for restoring the central insulin-mediated regulation of peripheral metabolism and physiological functions.

Among the factors that lead to the development of neurodegenerative processes and cognitive deficit in T2DM and MS, the most important are cerebrovascular dysfunctions [256,257], reduced glucose metabolism in brain structures [258], neuroinflammation [259,260,261,262,263], as well as central IR, which is also observed in patients with AD [258,262,264]. Impaired insulin signaling may affect only certain areas of the brain, which is due to region-specific changes in the expression, distribution, and activity of INSR and downstream components of the insulin signaling system [37,265,266,267]. As in the case of T2DM, patients with T1DM have impaired CNS function, which is referred to as “diabetic encephalopathy”. Its characteristic features are cerebral atrophy, lesions of the white matter of the brain, impaired synaptic plasticity, delayed latency of evoked potentials, as well as a decrease in the activity of a number of cognitive domains. The main causes of diabetic encephalopathy are acute insulin deficiency, severe hyperglycemia, hyperactivation of oxidative and inflammatory processes in the brain, as well as impaired blood circulation in the brain vessels, which differs in etiology and pathogenesis from that in T2DM [268].

In T2DM and MS, due to IR, the receptor-mediated transport of insulin across the BBB is weakened, which can lead to insulin deficiency in brain structures, despite normal or elevated insulin levels in the blood. A certain contribution to the disruption of such transport is made by the disintegration of the BBB, which occurs in conditions of DM2 and MS as a result of the activation of inflammatory and apoptotic processes, an increase in the production of reactive oxygen species, and a decrease in the sensitivity of the cells that form the BBB to insulin [65,269]. Therefore, under conditions of peripheral IR in T2DM, the weakening of insulin signaling in the brain, or at least in certain brain regions, may be due not only to central IR but also to the insufficient transfer of insulin from the blood to brain. We have shown that in rats and mice with hyperinsulinemia and severe IR, the level of insulin in the hypothalamus and other brain regions is reduced, which indicates the development of insulin deficiency in the CNS under conditions of systemic hyperinsulinemia [270,271]. Proteolytic enzymes, including the insulin-degrading enzyme, capable of cleaving the insulin molecule, can make a certain contribution to the regulation of insulin levels in the brain. At low insulin concentrations, the insulin-degrading enzyme hydrolyzes β-amyloid peptides with high efficiency, thereby preventing the formation of β-amyloid plaques and the AD development. However, under conditions of hyperinsulinemia in T2DM and MS, this enzyme begins to hydrolyze insulin, which, competing with β-amyloid peptides, promotes the accumulation of β-amyloid aggregates in cerebral vessels, which is the reason for the close relationship between T2DM and AD [272,273,274,275].

At present, studies of the effect of INI on functional disorders in DM can be divided into two groups. The first, most extensive group consists of experimental and clinical studies to study the restorative effect of INI on the functional activity of different brain regions, neuronal networks, and cognitive functions, and significant progress has been made in this regard. The second, relatively small group consists of works devoted to the ability of INI to improve hormonal and metabolic parameters in DM and restore the functions of peripheral organs and tissues [29,36,276].

The emphasis in the use of INI in patients with T2DM on the correction of cognitive deficit and the prevention of neurodegenerative changes is largely due to the fact that there is strong evidence that T2DM and AD are closely interconnected [277,278]. Thus, it has been established that the most important factors for the development of AD are prolonged hyperglycemia, decreased insulin sensitivity, increased inflammatory processes, including the development of neuroinflammation, and lipotoxicity, which are typical for patients with T2DM [278]. Since INI is widely used for the treatment of AD, it was assumed that insulin was highly effective in the treatment of elderly patients with T2DM and MS with signs of CNS damage, which was subsequently confirmed by clinical studies [29,279,280,281]. Importantly, both patients with T2DM have an increased risk of developing AD, and patients with AD and vascular dementia have a predisposition to T2DM [274,275,282]. That is, the etiology and pathogenesis of T2DM and AD are a two-way avenue. Thus, the treatment of one of these pathologies, including with the use of INI, can largely prevent the development of the other.

### 5.1. Clinical Studies

Currently, the final stages of clinical trials of INI for the treatment of elderly patients with T2DM are being carried out. The main attention is focused on the prevention of cognitive deficit and the consequences of CNS dysfunctions, while the effect of INI on metabolic and hormonal parameters is only at the initial stage of research. The results obtained in the clinic show that the basis of the restorative effect of INI on cognitive functions in diabetic patients is the improvement of blood flow in the brain regions [29,279,280] and the restoration of integrative relationships between the hippocampus and the brain’s default mode network, including the medial frontal cortex [38,280]. Functional magnetic resonance imaging has shown that even a single administration of INI (40 IU, Novolin) to elderly patients with T2DM restores impaired functional relationships between the default mode network and hippocampal regions, and after treatment with INI, the relationships in diabetic patients differed little from those in control subjects of the same age [280].

Gait and walking speed in the elderly are very informative markers of their mental and physical health. A decrease in walking speed, including in elderly patients with T2DM, positively correlates with impaired cerebral circulation (brain hypoperfusion) [283] and can be considered as a predictor of cognitive impairment and senile dementia [284]. Long-term, 24 weeks, treatment of elderly T2DM patients with INI (40 IU, Novolin) resulted in a significant improvement in both simple and dual-task walking speed, and this was due to improved verbal memory and CBF in the medial prefrontal cortex [34]. Walking speed and verbal memory also improved in older adults without T2DM who received INI in the same regimen [29]. In contrast to regular subcutaneous insulin administration in elderly patients with T2DM, long-term administration of INI did not cause hypoglycemic episodes and allowed better control of glucose levels. Along with this, in patients with 24-week administration of INI, a moderate decrease in blood insulin levels was found, which indicates in favor of an increase in tissue sensitivity to the hormone, as well as a trend towards a decrease in fasting glucose and glycated hemoglobin, indicating an improvement in glucose tolerance and partial restoration of glucose homeostasis [29,48,285].

As noted above, the ability to influence CBF is of decisive importance for the effects of brain insulin in the CNS. However, these effects are highly dependent on the region of the brain and the metabolic status of the patient. The mechanisms of insulin action on CBF are not fully understood, but the fact that INI affects the CBF velocity only in those brain regions where INSRs are present indicates the involvement of insulin signaling cascades in CBF regulation [264,266,267]. One of these mechanisms may be the effect of brain insulin on the activity of the neurovascular unit, including endothelial, neuronal, and glial cells. The neurovascular unit is involved in functional interaction with vascular cells, causing a change in their tone in response to activation or inhibition of neuronal activity [257,286]. Administration of INI to healthy individuals resulted in an increase in CBF velocity in the inferior frontal gyrus, dorsal striatum, and insular cortex [287], as well as a decrease in CBF velocity in the hypothalamus and around the middle frontal gyrus [267,288]. Since these brain areas are involved in the regulation of feeding behavior and the implementation of the reward and reinforcement program, in healthy subjects, INI affects appetite, motivation to food intake, and taste preferences, which is consistent with clinical observations [49,267,289,290]. At the same time, the effect of INI on hunger, perception of food stimuli, and reward behavior in women and men differs significantly, indicating a modulating effect of gender factor on INI-induced feeding behavior [290]. The multidirectional effects of INI on the CBF in various brain regions leads to the fact that a significant effect of INI on the global CBF in healthy subjects has not yet been found.

The effect of INI on the CBF velocity in the brain of patients with T2DM differs significantly from that in healthy individuals [257]. In T2DM patients, a single treatment with INI (40 IU) affected the CBF velocity only in the insular cortex, causing its increase. At the same time, this effect was significantly more pronounced in diabetic patients than in healthy subjects [279]. The degree of vasodilatation in the middle cerebral artery territory and improvement of regional blood flow in the insular zone in patients with T2DM positively correlated with the restoration of visuospatial memory, and in healthy subjects, with the verbal fluency test [279]. It should be noted that dysfunctions of the insular cortex and disintegration of its subregions in patients with T2DM negatively affect their emotional, cognitive and sensorimotor functions [291]. Treatment of patients with T2DM with INI did not reveal noticeable changes in CBF in other brain regions, including the hypothalamus and amygdala, which are associated with peripheral IR. At the same time, when studying the effect of INI on patients with reduced insulin sensitivity, but without overt T2DM, a significant change in CBF velocity in these areas of the brain was shown [292,293,294]. However, in studies with non-diabetic patients, doses of INI that caused a decrease in CBF in the hypothalamus [292,293] and amygdala [294] were 160 IU and were many times higher than the doses used in the study by Vera Novak and coauthors [279]. Moreover, according to some researchers, in order to detect significant changes in the functional activity of the brain, including regional blood flow, relatively high doses of INI (80 IU or more) are needed [288], while other scientists indicate that INI doses of 40 IU are effective [287]. It has been shown that INI at a dose of 40 IU causes specific changes in certain brain regions responsible for feeding behavior, reward reactions, and visuospatial memory [287]. Thus, further studies are needed to clearly show in which areas of the brain and under what modes of INI use the CBF velocity changes and how these changes are associated with an improvement in cognitive functions and peripheral insulin sensitivity [257]. It is also important that INI doses of 160 IU and above can cause a number of side effects in patients with T2DM [295]. Along with this, the combination of physical activity and the use of INI can lead to a change in the safe dose of INI. Insulin doses of 100 IU and above increased the risk of hypoglycemic episodes in students who performed moderate exercise on an empty stomach [47].

### 5.2. Experimental Studies

There are relatively few experimental studies on the effectiveness of INI in the correction of cognitive, metabolic and functional disorders caused by T2DM. This is largely due to the fact that clinical studies of the effectiveness of INI in T2DM paradoxically outstripped experimental works. So, already at the initial period, encouraging results were obtained demonstrating the ability of INI to improve cognitive functions in patients with T2DM, and the safety of its use was also shown. However, animal models are useful and indispensable for establishing the molecular mechanisms and targets of the neuroprotective action of INI in DM, as well as for studying INI effects on metabolism, hormonal parameters, and functions of the peripheral organs and tissues in diabetic pathology. Along with this, studies of the effectiveness of INI in models with insulin-deficient forms of DM, such as T1DM and decompensated severe T2DM, are of interest, since clinical data on the effectiveness of INI in insulin-deficient DM are not yet available.

We have shown that administration of low doses of INI (0.25–1.5 IU/rat) to rats with models of T1DM, T2DM, and MS, with different durations, leads to an improvement in their metabolic and hormonal parameters and improved cognitive functions, and in the case of severe form of T1DM, such treatment increases the survival of animals [206,208,209,210,296,297,298,299,300,301,302]. Treatment of diabetic rats with INI moderately reduced hyperglycemia without causing hypoglycemic episodes, attenuated hyperphagia, and in the case of T2DM and diet-induced MS, INI increased tissue sensitivity to insulin and leptin and normalized glucose tolerance [207,210,302]. These effects were associated with partial restoration in the hypothalamic signaling pathways regulated by peptide neurohormones and biogenic amines (serotonin and dopamine), as well as with inhibition of apoptotic processes in the brain [209,301,302]. One of the results of this was the partial restoration of spatial memory and learning ability, as we demonstrated in severe streptozotocin DM and in insulin-deficient “neonatal” model of T2DM, caused by the treatment of neonatal rat pups with high-dose streptozotocin [296,300].

One of the promising approaches for the correction of brain functions in DM and MS is the combined use of INI with intranasally administered proinsulin C-peptide (ICP). C-peptide, synthesized in pancreatic beta cells from proinsulin, is the most important regulator of the nervous, cardiovascular, endocrine, and other systems [303,304,305]. The main mechanism of C-peptide action is considered to be its ability to form hetero-oligomeric complexes with insulin, which increases insulin bioavailability and potentiates its stimulatory effects on the insulin signaling system [306,307,308]. At the same time, the ability of the C-peptide to bind to its specific receptors and, thus, independently activate intracellular effector systems cannot be excluded [309,310]. Under conditions of T1DM, the production of not only insulin but also C-peptide decreases significantly, which leads to a deficiency of C-peptide in the CNS and is not compensated by insulin monotherapy. In accordance with this, the combined use of INI and ICP, taken in a certain ratio that is optimal for the effective formation of heterooligomeric complexes, can become a promising approach for correcting the deficiency of the insulin signaling (and, possibly, the C-peptide signaling) in the brain in T1DM. This is evidenced by the results of our studies in rats with streptozotocin DM, demonstrating an increase in the effectiveness of INI when combined with ICP in the molar ratios of 1:1–1:3 [208,301,302,311].

The nine-day treatment of male Wistar rats with long-term moderate T1DM induced by 35 mg/kg streptozotocin and seven-day treatment of animals with short-term severe T1DM induced by 50 mg/kg streptozotocin with a mixture of INI (20 µg/rat) and ICP (36 µg/rat) attenuated hyperglycemia and restored reduced leptin levels. In the hypothalamus, such treatment enhanced the reduced expression of genes encoding D1- and D2-dopamine receptors and MC3- and MC4-melanocortin receptors involved in the regulation of the nervous and endocrine systems [302]. We also observed normalization of the hypothalamic expression of the mitofusin-2 and Beclin-1 genes that control mitochondrial fusion and autophagy, as well as the ratio of mitofusins-1 and -2, which indicates the restoration of mitochondrial dynamics and the weakening of apoptosis [301]. The restorative effects of the combined use of INI and ICP on various parameters were superior to those of INI monotherapy and were more pronounced in moderate T1DM. Paradoxically, in the treatment of T1DM rats, the combination of INI and ICP, in contrast to INI monotherapy, did not cause an increase in the gene expression of phosphatase PTP1B, a negative regulator of insulin signaling, which can be considered as one of the mechanisms of the ICP-induced potentiation of the stimulating effects of INI on the brain insulin system [311]. Along with T1DM, the combination of INI and ICP was also effective in insulin-deficient “neonatal” model of T2DM, and in this case, along with an improvement in metabolic parameters, a partial restoration of the sensitivity of the adenylate cyclase signaling system to hormonal regulation in the hypothalamus, myocardium, and adipose tissue was shown [207].

The ability of insulin to normalize mitochondrial function in the brain, impaired in DM, can make a significant contribution to the neuroprotective effect of INI in diabetic pathology [312,313]. Treatment of male rats with streptozotocin DM with a relatively high dose of INI (10 IU/rat) for 11 days led to a significant restoration of complex I and IV of the mitochondrial respiratory chain in the brain and a decrease in the production of reactive oxygen species, and also improved the functional activity of mitochondrial ATP-sensitive large-conductance Ca^2+^-activated potassium (mitoBKCa) channels and increased expression of their β-subunit [313]. Our data on INI-induced normalization of the expression of genes involved in mitochondrial biogenesis and apoptosis in the hypothalamus of rats with T1DM support the normalizing effect of INI on mitochondrial function in DM [301].

Thus, experimental studies not only confirm the effectiveness of INI demonstrated in patients with T2DM but also allow us to evaluate some of the molecular mechanisms of the restorative effects of INI in DM, expand the spectrum of its action to the periphery, and indicate its possible therapeutic potential in T1DM and severe forms of T2DM. In addition, they show promise for the use of INI in combination with other insulin signaling regulators, as shown for the combination of INI and ICP. In this regard, it should be noted that the restorative effects of INI on glucose homeostasis are enhanced when it is combined with metformin in rats with MS [314].

## 6. Intranasal Insulin and Anesthesia

Cognitive disorders are one of the most common postoperative complications. Postoperative delirium (POD) develops as early as 5–7 days after surgery, which takes place with the use of anesthesia. Later, POD leads to cognitive impairment, which significantly increases the duration of treatment and hospitalization, worsens the prognosis of surgical intervention, and increases its mortality [315]. Despite the fact that the pathophysiology of POD is not well understood, it is believed that its main mechanisms are impaired neuroendocrine regulation, neuroinflammation, neural network dysconnectivity, as well as ischemic damage caused by a decrease in CBF [316]. The consequence of postoperative ischemia is severe metabolic disorders in the brain tissue due to the inhibition of aerobic processes, which is illustrated by an increase in the level of lactate in the cerebrospinal fluid and a decrease in the activity of neuron-specific enolase [315,317,318,319].

Cognitive disorders that develop after POD are divided into delayed neurocognitive recovery (DNR), which is due to neurocognitive impairment observed in patients from the 7th to the 30th day after surgery, and postoperative neurocognitive disorder (pNCD), which is observed from the 30th day to one year after surgery [320]. The frequency of DNR and pNCD depends on many factors, such as age; type of surgical intervention; comorbidities (DM, respiratory diseases, etc.); psychophysiological state of the patient; and perioperative hyperglycemia, as well as tests used to assess cognitive disorders [320,321,322,323]. For the development of DNR and pNCD, the type and depth of anesthesia are important, and among the pathophysiological mechanisms that determine the relationship between postoperative cognitive disorders and anesthesia, it is necessary to highlight mitochondrial dysfunctions, increased inflammatory and apoptotic processes in neurons, hyperphosphorylation of the tau protein, and also the accumulation of β-amyloid proteins and the formation of their aggregates [324,325,326,327].

### 6.1. Experimental Studies

The results of experimental studies show that INI prevents anesthesia-induced DNR and pNCD, and also prevents the development of neurodegenerative changes in brain structures, the activation of apoptotic processes in them, and neuroinflammation. It is important that the neuroprotective effects of INI are realized with various schemes of its use and with various forms of anesthesia. In 2014, Chinese scientists found that the treatment of transgenic 3xTg-AD mice predisposed to AD with INI (1.75 IU/day) for 7 days prevents hyperphosphorylation of the tau protein and the triggering of neurodegenerative changes in the brain caused by anesthesia with propofol (i.p., 250 mg/kg) [328]. This effect was due to an INI-induced increase in the activity of protein phosphatase 2A, which is involved in the dephosphorylation of the tau protein, as well as inhibition of the activity of a number of kinases, such as cyclin-dependent protein kinase 5, calcium/calmodulin-dependent protein kinase II, and c-Jun N-terminal kinase, which are responsible for the phosphorylation of the tau protein [328]. A reduction in anesthesia-induced tau hyperphosphorylation has been demonstrated in wild-type mice treated with INI following the same protocol, in which case recovery of post-anesthetic impaired spatial memory and learning ability as assessed by the Morris water test was demonstrated [329]. Seven-day INI treatment of aging mice prevented anesthesia-induced tau hyperphosphorylation via the 3-phosphoinositide cascade, whose activation by insulin resulted in inhibition of GSK-3β activity involved in tau protein phosphorylation [330]. After anesthesia, INI also restored the decreased expression of a number of synaptic proteins and brain-derived neurotrophic factor (BDNF), which are involved in neuronal survival and synaptic brain plasticity [331]. INI-induced normalization of the tau protein phosphorylation and the expression of synaptic proteins and BDNF in the brain were positively correlated with improved cognitive functions in anesthetized animals [330,331].

The recovery of cognitive functions and a pronounced neuroprotective effect of INI (1.75 IU/day) were also shown in 3xTg-AD and wild-type mice treated with INI for three days prior to anesthesia, which was achieved with propofol treatment followed by an hour-long inhalation of sevoflurane. The most pronounced restorative effect of INI on anesthesia-induced cognitive dysfunctions was observed in 3xTg-AD mice, as well as in aging, but not young, wild-type mice [332]. Other authors have shown that treatment of mice for three days with INI (0.14 IU/animal, 30 min before the start of anesthesia with sevoflurane on days 7, 8, and 9 of postnatal development) prevented cognitive impairment and behavioral abnormalities, developing later in adulthood (3–4 months) and elderly (16–18 months) age [323,324]. In the brain, INI also restored the expression of the adapter protein PSD95 (postsynaptic density 95), which is responsible for the clustering of ion channels on the surface of the postsynaptic membrane, and normalized the increased after anesthesia activity of proapoptotic caspase pathways [333,334]. These studies are important because a significant proportion of children who were repeatedly anesthetized as infants later developed learning disabilities and a range of other cognitive dysfunctions [335,336,337]. This is due to the fact that sevoflurane, often used for anesthesia, such as other anesthetics, is characterized by potential neurotoxicity for the developing brain [338,339].

Mitochondrial dysfunctions, hyperproduction of reactive oxygen species, and activation of mitochondrial apoptosis, as demonstrated in the case of anesthesia in juvenile animals, are one of the factors of CNS dysfunction during early anesthesia [340,341]. These disorders may be due to impaired activity of the signaling pathway, including the mTORC1 (Figure 1). This complex, which is a target for many regulatory factors, including insulin and IGF-1, controls the functions and biogenesis of mitochondria and, as a result, is responsible for the energy metabolism of the cell. This is based on mTORC1-mediated regulation of RNA translation, which depends on such mTORC1 targets as eukaryotic translation initiation factor 4E (eIF4E)-binding protein (4E-BP) and ribosomal protein S6 kinase [342,343]. The following facts indicate the role of the mTORC1 and its targets in the development of neurodegenerative disorders and cognitive deficits caused by anesthesia. Knockout of the gene encoding the repressor of the mTORC1-dependent translation 4E-BP in juvenile mice significantly attenuates the negative effects of three-day (P15–P17) anesthesia with isoflurane (1.5%, 2 h), preventing a decrease in the survival of hippocampal cells and impaired neuronal plasticity [344]. Administration to mice for 7 days (P8–P14) of INI (2 IU/animal/day, Humulin R), a potent mTORC1 activator, prior to anesthesia also prevents hippocampal cell death, while treatment of animals with an mTORC1 inhibitor completely abolishes the restorative effects of INI on neuronal plasticity [344]. Thus, insulin-mediated activation of the PI3K/AKT/mTORC1/4E-BP2 signaling cascade in the hippocampus of anesthetized mice may be one of the most important neuroprotective mechanisms of INI, especially at the early stages of brain formation.

The experimental results presented above clearly demonstrate that pre-treatment of animals with INI prior to anesthesia significantly attenuates or even prevents neurodegeneration and decreased neuronal plasticity in the brain, thereby preventing the development of DNR and pNCD. It should also be noted that the spectrum of cognitive impairment induced by anesthesia and the potential mechanisms that cause them are in many respects similar to those functional changes in the CNS that are observed in AD, Parkinson’s disease, mild cognitive impairment, and also in DM, in which the effectiveness of INI therapy in can now be considered proven [327]. Based on this, there is every reason to believe that pre-treatment of patients with INI before the anesthesia procedure can have a significant neuroprotective effect and help avoid many postoperative complications.

### 6.2. Clinical Study

The results of a clinical trial of INI for the treatment of POD in elderly patients have now been published [345], and a clinical trial of the use of INI for the treatment of POD in patients over 65 years of age hospitalized in geriatric medicine has been announced [346]. In 2021, a randomized, placebo-controlled trial examined the effects of INI (20 IU) when given 2 days prior to laparoscopic radical gastrointestinal surgery [345]. The study involved 80 patients. The INI-treated group showed a decrease in the proportion of POD within 5 days after surgery. These patients showed a decrease in inflammatory markers such as TNF-α, IL-1β and IL-6. There were no changes in blood glucose levels or other side effects when patients were treated with INI, which indicates the safety of such therapy [345]. Along with this, a randomized, placebo-controlled, double-blind study was recently launched with a 6-month follow-up of patients with signs of POD [346]. The efficacy of long-term treatment of patients with INI (Detemir, 20 IU twice daily) until the clinical manifestations of POD will be studied. The authors aim to find out whether INI therapy can reduce the severity of delirium, reduce the number of hospital complications and the number of new hospitalizations, reduce or eliminate the use of psychotropic drugs during hospital stay, and reduce mortality.

## 7. Intranasal Insulin and the Gonadal and Thyroid Systems

In addition to controlling metabolic processes and functioning as a neuroprotector, insulin, such as other peptides of the insulin group, plays an important role in the regulation of the functioning of the neuroendocrine and endocrine systems. This is due to the fact that both peripheral and central links of the hypothalamic-pituitary system are targets for insulin, since they contain all the main components of the insulin signaling system [347,348].

### 7.1. Gonadal Axis

INSR, as well as the other components of the insulin system, are present in the testes and ovaries, and from the embryonic stage [347,349,350], and this indicates participation of insulin not only in the regulation of reproductive functions but also in gonadogenesis and sex-specific development [347,348]. It is interesting to note that small amounts of insulin are synthesized in the testes, which indicates its inherent functions as a testicular auto- and paracrine regulator [351]. The most important targets for insulin in the brain are the hypothalamic neurons responsible for the production of gonadoliberin (GnRH), releasing factor for luteinizing (LH) and follicle-stimulating hormones (Figure 3). The stimulating effect of insulin on them is indirect, being realized through at least three mechanisms: (1) through the activation of kisspeptin-expressing neurons that secrete the neurohormone kisspeptin, an activator of GnRH production, (2) through the activation of neurons expressing pro-opiomelanocortin (POMC), a precursor of melanocortin peptides that either directly or indirectly, through activation of kisspeptin-expressing neurons, increase GnRH release, or (3) by inhibition of hypothalamic neurons expressing agouti-related peptide (AgRP) and neuropeptide Y (NPY), which negatively affect GnRH synthesis [352,353]. INSR and other components of insulin signaling are present in all of these populations of hypothalamic neurons, although they are not found in GnRH-expressing neurons. It is important that the adipokine leptin uses the same mechanisms to stimulate GnRH production, being an insulin synergist in the activation of the hypothalamic links of the gonadal axis [354,355].

In metabolic disorders, an INI-induced increase in insulin levels and an improvement in the insulin signaling system in the hypothalamus lead to the restoration of mitochondrial functions, an increase in the survival of hypothalamic neurons, and a decrease in oxidative stress and neuroinflammation. All this makes a significant contribution to the improvement of the functions of the hypothalamic links of the gonadal and thyroid axes. Along with this, INI through various mechanisms can activate hypothalamic neurons expressing gonadoliberin (GnRH) and thyroliberin (TRH), which leads to an increase in GnRH and TRH secretion. In conditions of DM and MS, this leads to normalization of the levels of pituitary hormones (gonadotropins, TSH) and sex steroid and thyroid hormones in the blood. The result of this is the restoration of reproductive functions, including spermatogenesis, folliculogenesis, and oogenesis, as well as the physiological processes regulated by thyroid hormones. The possibility of non-hypothalamic mechanisms of the restorative effect of INI on the gonads and the thyroid through INI-induced improvement in metabolism and physiological functions cannot be ruled out, but this needs to be studied.

It should be emphasized that in terms of importance for sex determination and reproduction, insulin is largely inferior to IGF-1 and IGF-2, the members of the insulin peptides family, which are key regulators of embryonic gonadal development, spermatogenesis, folliculogenesis, and testicular and ovarian steroidogenesis, as well as affect the processes of ovulation and embryogenesis [356]. The recognition of the key role of IGF-1 and IGF-1-activated signaling systems in the regulation of reproduction is evidenced by the recently developed concept of combining the hypothalamic-pituitary-gonadal and hypothalamic-pituitary-somatotropic axes into a common somatotropic-testicular/ovarian axis [357,358]. At the same time, since insulin is capable of activating hybrid INSR/IGF1R receptors, and also has common links with IGF-1/IGF-2-activated intracellular cascades, many effects of insulin on the gonadal axis can be realized in close interaction with IGF-1-system or even be mediated by it.

Functional disorders of insulin signaling in DM, obesity, and other metabolic disorders can be one of the causes of reproductive dysfunctions and reduced fertility [359,360]. It has been established that IR and hyperinsulinemia are the most important factors leading to the development of polycystic ovary syndrome, a severe endocrine disorder in women inducing to infertility [361]. Changes in insulin signaling, caused by both IR and insulin deficiency, lead to dysfunctions in the gonadal axis both at the level of peripheral (gonads) and upstream links (hypothalamus, pituitary), resulting in dysregulation of the gonadal axis [360]. Normalization of insulin levels and insulin signaling in the brain, including using INI, may be one of the promising approaches to restore the activity of the gonadal axis in metabolic disorders. In this regard, it should be noted that there is evidence of a restorative effect of an intranasally administered leptin fragment on testicular steroidogenesis and spermatogenesis in mice with heat-induced testicular dysfunction [362]. However, data on the effect of INI on steroidogenesis and reproductive functions in metabolic disorders are very scarce.

We have shown that nine-day treatment of male rats with mild T1DM induced by a moderate dose of streptozotocin (35 mg/kg) with INI (20 μg/rat/day) and its combination with ICP (36 µg/rat/day) partially restored their androgenic status, and the combination of INI and ICP was superior to INI monotherapy [208]. It is important to note that the increase in testosterone levels was not accompanied by an increase in the blood LH level, which changed little both in mild T1DM and in diabetic animals treated with INI and the combination of INI and ICP. Thus, the normalization of testosterone levels may be associated not with the activation of the upstream links of the gonadal axis but with an increase in the testicular response to LH. This is supported by our data on an increase in the stimulatory effect of human chorionic gonadotropin (hCG) on the activity of the adenylate cyclase system in the testicular membranes of rats treated with INI and ICP as compared to untreated diabetic animals [208]. The same treatment regimen was less effective in male rats with severe T1DM induced by a high dose of streptozotocin (65 mg/kg), with their characteristic acute androgen deficiency and a significant decrease in blood LH level [302]. In this case, in the groups treated with INI and the combination of INI and ICP, an almost two-fold increase in the blood level of LH (to that in the control) was shown, but there was no significant increase in testosterone production, which, we believe, is due to functional disorders in the testes, leading to a critical decrease in their sensitivity to gonadotropins [302]. It has been shown that INI effectively restored metabolic and hormonal disorders that developed in adulthood in male rats with interruption of breastfeeding in the early postnatal period (P19–P21) [363]. The treatment of male rats with INI, both in the postnatal period (P28–P55, 0.25 IU/rat/day) and in adulthood (P183–P210, 0.5 IU/rat/day), led to the normalization of blood testosterone levels, reduced in untreated animals, and also improved the response to GnRH, indicating an increase in the sensitivity of the testes to gonadotropins [363]. We have also shown that the sensitivity of the adenylate cyclase signaling system to gonadotropins significantly improved in the ovarian membranes of female rats with the “neonatal” model of T2DM, which were treated with INI for a long time [298]. These results indicate the prospects for the use of INI to normalize the functions of the gonadal axis in moderate T1DM and in metabolic diseases with IR and hyperinsulinemia, while, in severe forms of T1DM, this treatment is not as effective or requires higher doses of INI and longer duration.

### 7.2. Thyroidal Axis

As with the gonadal axis, insulin in the hypothalamus is able to stimulate the synthesis and secretion of thyroliberin (TRH), the thyroid stimulating hormone (TSH) releasing factor, by activating POMC-expressing neurons and by inhibiting AgRP/NPY-expressing neurons [364,365] (Figure 3). Melanocortin peptides generated from POMC stimulate TRH release, while AgRP and NPY, on the contrary, suppress it [366,367,368,369]. The functions of the thyroid axis change not only due to autoimmune thyroid diseases but also in metabolic disorders, including T1DM, T2DM, and MS [370,371], and autoimmune thyroid diseases are closely interrelated with diabetic pathology [372]. One of the causes of thyroid dysfunction in DM and MS may be the weakening of insulin signaling in the brain and the disruption of the interaction between the hypothalamic insulin and leptin pathways, which are synergistic to stimulate TRH production. It has been established that leptin, such as insulin, stimulates TRH release by activating POMC-expressing neurons [368] and inhibiting AgRP/NPY-expressing neurons [369]. The weakening of hypothalamic insulin and leptin signaling as a result of inactivation of IRS4, which couples the insulin and leptin receptors with PI3K, leads to a decrease in TRH production and central hypothyroidism [373]. Thus, in hypothyroid patients with DM and MS, as well as in the correction of thyroid hormone deficiency induced by insufficient TRH secretion, the use of INI may be useful.

We studied the effect of four-week treatment of rats with severe T1DM with different doses of INI (0.3, 0.6 and 1.5 IU/rat/day) and long-term treatment of rats with mild T1DM for 135 days with INI at a dose of 0.45 IU/rat /day. Such treatment resulted in improved thyroid function, as indicated by an increase in the level of thyroid hormones and restoration of the response to TRH, which was decreased in T1DM [206]. Long-term (135 days) administration of INI to control (non-diabetic) rats and animals with a mild T1DM significantly increased the plasma level of TSH. Thus, long-term treatment of rats with T1DM using even relatively low doses of INI, on the one hand, leads to the restoration of the functions of the hypothalamic-pituitary-thyroid axis and, on the other hand, can cause overproduction of TSH [206]. The TSH-stimulating effect of INI should be taken into account not only when developing a strategy for using INI to correct metabolic and endocrine disorders but also in the treatment of neurodegenerative diseases, including AD and Parkinson’s disease. In rats with different models of T1DM, we also demonstrated the restorative effect of INI monotherapy and the combination of INI with ICP on thyroid hormone levels and response to thyroliberin reduced in diabetes [208,302]. Along with this, INI normalized the level of thyroid hormones in adult rats with MS, which was induced by a three-day interruption of breastfeeding (P19–P21 days) [363]. These results point to the prospects for the use of INI and its combinations for the treatment of certain types of thyroid pathology, which requires further systemic research.

## 8. Conclusions

The study of insulin and its system in the CNS indicates the role of brain insulin as an important regulator of growth, differentiation, and survival of neurons and glial cells, autophagy, neuronal plasticity, CBF, and nerve cell metabolism. Experimental models have shown that impaired insulin signaling inevitably leads to neurodegeneration, while the insulin and activators of insulin signaling, on the contrary, increase the survival of nerve cells and prevent neurodegenerative changes. All this has led to the fact that for many years the main attention of researchers has been focused on the clinical use of INI for the treatment and prevention of neurodegenerative diseases. The logical result of this was the successful experience of using INI for the treatment of AD and mild cognitive impairment. It should be emphasized that AD is characterized by close relationships of neurodegenerative changes with the development of central IR and impaired glucose homeostasis in the brain, which is why it is often called “type 3 diabetes”. Along with the treatment of neurodegenerative diseases, considerable attention has been paid to improving cognitive abilities, improving mood and reducing anxiety in case of excessive prolonged mental and physical activity, fatigue, apathy, and chronic stress in healthy subjects. Accordingly, those diseases or conditions that were not directly related to neurodegeneration or to transient impairment of attention and cognitive functions, as a rule, were not the subject of research on the potential effectiveness of INI. Moreover, in accordance with the “anti-neurodegenerative” paradigm, for a long time, the main goal in the treatment of patients with DM with INI was the prevention or reduction of DM-associated cognitive deficit, while other possible therapeutic effects of INI were not in the focus of attention.

At the same time, the recent studies have shown that the therapeutic potential, as well as the range of possible targets of INI, is much wider, which is largely due to new data on the regulatory effect of brain insulin on metabolic processes and energy consumption at the periphery, as well as on the functions of the endocrine systems. This is due to the pleiotropic action of insulin, which is due to the great diversity of its intracellular signaling cascades and their crosstalk with other signaling cascades regulated by adipokines, growth factors, and biogenic amines, as well as the ubiquitous distribution of INSR and other components of the insulin system in organs and tissues, including all brain regions. The presence of INSR/IGF1R hybrid receptors enables insulin to influence IGF-1-activated signaling pathways, which also have many targets and are widely distributed. Despite the fact that INI only slightly affects glucose homeostasis in DM, its ability to increase insulin sensitivity and thereby alleviate hyperglycemia is gaining new evidence. INI reduces appetite and thus attenuates metabolic and hormonal disfunctions in patients with obesity and MS, including the attenuation of peripheral IR. Experimental evidence has been obtained for the ability of INI to restore the functions of the gonadal and thyroidal axes in rodents with DM and MS that are deficient in sex steroid and thyroid hormones. All this significantly expands the possible pattern of using INI for the treatment of diabetic pathology, making it possible to go far beyond the correction of exclusively diabetic encephalopathy.

A new and little developed direction is the use of INI to correct damage caused by cerebral ischemia and reperfusion, brain trauma, as well as anesthesia leading to postoperative delirium and a number of other severe complications. And here it is important to adequately determine and further optimize the therapeutic window for the use of INI, which would allow to restore damaged brain structures or even prevent neurodegenerative processes. Fundamental, especially for anesthesia, is to determine when, in what doses and in what mode INI should be used to achieve the greatest therapeutic effect. The combined use of INI with proinsulin C-peptide, which is able to modulate the regulatory effects of insulin, as well as with functional insulin synergists, such as leptin, glucagon-like peptide-1, melatonin, and different neuroprotectors, may become promising. All this requires further research, and every year this new direction of application of INI in medicine becomes an increasingly intriguing and exciting task for specialists in the field of neurobiology, endocrinology and molecular medicine.

## Figures and Tables

**Figure 1 ijms-24-03278-f001:**
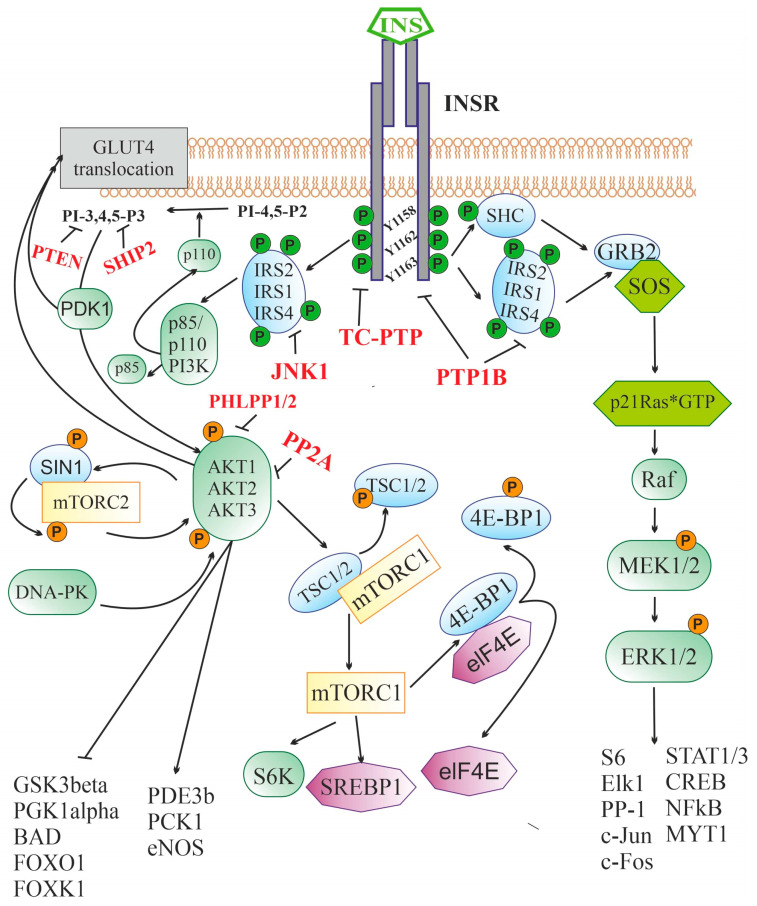
Insulin-activated signaling pathways. Abbreviations: INSR—insulin receptor, containing three sites for tyrosine phosphorylation (Y1158, Y1162, and Y1163); IRS1, IRS2, and IRS4—insulin receptor substrates-1, -2 and -4, respectively; p85 and p110 PI3K—p85-regulatory and p110-catalytic subunits of phosphatidylinositol-3-kinase; PI-4,5-P2—phosphatidylinositol-4,5-diphosphate; PI-3,4,5-P3—phosphatidylinositol-3,4,5-triphosphate; PDK1—phosphoinositol-dependent protein kinase-1; AKT1, AKT2, and AKT3—serine/threonine-specific protein kinase B (AKT kinase) of the types 1, 2 and 3, respectively; DNA-PK—DNA-dependent protein kinase; mTORC2—mammalian target of rapamycin complex 2 that includes the protein kinase mTOR, the regulatory protein RICTOR (Companion of mammalian Target Of Rapamycin), mSIN1 protein (mammalian Stress-activated protein kinase Interacting Protein 1) and some other components; GLUT4—type 4 insulin-dependent glucose transporter; GSK3beta—glycogen synthase kinase-3β; PGK1alpha—peroxisome proliferator-activated receptor-gamma coactivator 1α; BAD—BCL2 antagonist in cell death; FOXO1 and FOXK1—transcription factors of FOX (forkhead box) family; PDE3B—subtype 3B cAMP-dependent phosphodiesterase; PCK1—phosphoenolpyruvate carboxykinase 1; eNOS—endothelial isoform of NO synthase; mTORC1—mammalian target of rapamycin complex 1; TSC1/2—tuberous sclerosis proteins 1 (hamartin) and 2 (tuberin); S6K—p70 ribosomal S6 kinase; SREBP1—Sterol regulatory element-binding transcription factor 1; eIF-4E—eukaryotic translation initiation factor 4E; 4E-BP1—eukaryotic translation initiation factor 4E-binding protein 1; SHC—adapter SH2/α-collagen-like protein; GRB2—adapter protein-2 associated with growth factor receptors; SOS—metabolic protein (Son of Sevenless), induced GDP/GTP exchange; p21Ras*GTP—small GTP-binding protein of the Ras family in GTP-bound form; Raf—serine/threonine-specific protein kinase; MEK1/2—mitogen-activated protein kinase kinases-1 and -2; ERK1/2—mitogen-activated protein kinases-1 and -2; Elk-1—transcription factor containing the ETS (E26 transformation-specific) domain; STAT1/3—types 1 and 3 signal transducers and transcription activators; CREB—cAMP-dependent transcription factor; S6—ribosomal protein S6; NF-κB—nuclear factor κ-light-chain-enhancer of activated B cells; MYT1—myelin transcription factor 1; c-Fos and c-Jun—the transcription factors of the Fos and Jun families, respectively; PP-1—protein phosphatase 1; PTEN—phosphatase (Phosphatase and TENsin homolog) hydrolyzing PI-3,4,5-P3 to PI-4,5-P2; SHIP-2—SH2 domain-containing inositol-5’-phosphatase 2 hydrolyzing PI-3,4,5-P3 to PI-3,4-P2; JNK1—c-Jun N-terminal kinase-1; PP2A—protein phosphatase 2A subtype; PHLPP1/2—plekstrin-homologous (PH)-domain leucine-rich-repeat-containing protein phosphatases; PTP1B—protein phosphotyrosine phosphatase 1B; TC-PTP—T-cell protein phosphotyrosine phosphatase.

**Figure 2 ijms-24-03278-f002:**
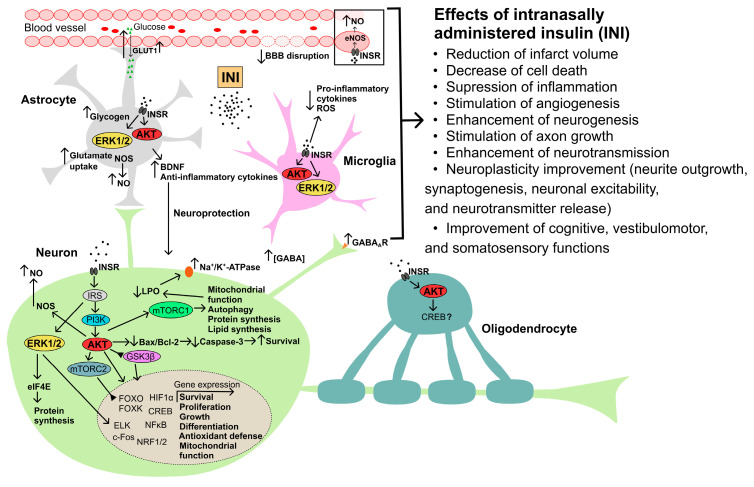
The possible mechanisms of neuroprotective action of intranasally administered insulin in cerebral ischemia and traumatic brain injury. Abbreviations: AKT—AKT kinase; Bax—Bcl-2-associated X protein, proapoptotic factor; BBB—blood–brain barrier; Bcl-2—B-cell lymphoma 2 protein, antiapoptotic factor; BDNF—brain-derived neurotrophic factor; c-Fos—transcription factor of the FOS family; CREB—cAMP response element-binding protein; eIF4E—eukaryotic initiation factor 4E; ELK—ETS Like-1 protein Elk; ERK1/2—extracellular signal-regulated kinases-1 and -2; FOXO and FOXK—forkhead box transcription factors of the O and K subfamilies; GABA—γ-aminobutyric acid; GABAAR—ionotropic GABAA receptor; GLUT1—type 1 glucose transporter; GSK3β—glycogen synthase kinase-3β; HIF1α—hypoxia-induced factor-1α; INSR—insulin receptor; IRS—insulin receptor substrate; LPO—lipid peroxidation; mTORC1, mTORC2—mammalian target of rapamycin complexes 1 and 2, respectively; NF-κB—nuclear factor kappa-light-chain-enhancer of activated B cells; NO—nitric oxide; NOS—NO synthase; NRF1/2—nuclear respiratory factors 1 and 2; ROS—radical oxygen species.

**Figure 3 ijms-24-03278-f003:**
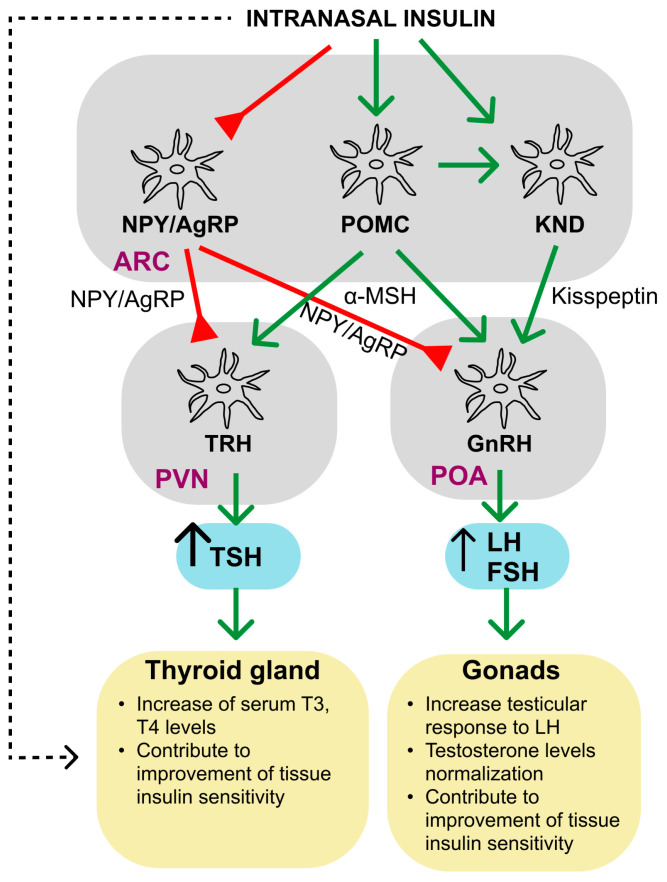
Proposed mechanisms and results of the action of INI in diabetes and metabolic syndrome on the hypothalamic links of the gonadal and thyroid axes. Abbreviations: AgRP—agouti-related peptide; ARC—arcuate nucleus of hypothalamus; FSH—follicle-stimulating hormone; GnRH—gonadotropin-releasing hormone; KND—hypothalamic neurons expressing kisspeptin, neurokinin B, and dynorphin; LH—luteinizing hormone; α-MSH—α-melanocyte stimulating hormone; NPY—neuropeptide Y; PVN—paraventricular nucleus of hypothalamus; POA—preoptic area of hypothalamus; POMC—pro-opiomelanocortin; T3, T4—thyroid hormones, triiodothyronine (T3) and thyroxine (T4), respectively; TRH—thyrotropin-releasing hormone; TSH—thyroid-stimulating hormone.

## Data Availability

Not applicable.

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
