# Peer review of "Hot Spots for the Use of Intranasal Insulin: Cerebral Ischemia, Brain Injury, Diabetes Mellitus, Endocrine Disorders and Postoperative Delirium"

_ijms, 2023, doi:10.3390/ijms24043278_

Round 1

Reviewer 1 Report

 “This review is devoted to the prospects and current trends in the use of INI for the treatment of these diseases, which, although differing in etiology and pathogenesis, are characterized by impaired insulin signaling in the brain.”  This review represents a lot of hard work by the authors and covers many interesting topics related to the prospects and current trends in the use of INI.  I hope that my comments and suggestions will help the authors to improve their manuscript.

The Abstract of this manuscript states the following:

1.      “A decrease in the activity of the insulin signaling system of the brain, due to both central insulin resistance and insulin deficiency, leads to neurodegeneration and impaired regulation of appetite, metabolism, endocrine functions.  This is due to the neuroprotective properties of brain insulin and its leading role in the functioning of brain signaling network responsible for the regulation of the nervous, endocrine and other systems.”

What the Abstract does not explicitly briefly state is the following: glucose metabolism is the main source of brain cell energy, glucose hypometabolism occurs in the brain in numerous neurodegenerative diseases, including Alzheimer’s and Parkinson’s resulting in a loss of brain cell energy and that intranasal insulin plays a leading role in stimulating glucose uptake and metabolism which increases the production of brain cell energy in the form of ATP and Phosphocreatine.  [You should mention and reference somewhere in your manuscript the following paper: Jauch-Chara K, et al. Intranasal insulin suppresses food intake via enhancement of brain energy levels in humans. Diabetes. 2012 Sep;61(9):2261-8. doi: 10.2337/db12-0025. Epub 2012 May 14. PMID: 22586589; PMCID: PMC3425399.  (I suggest you consider briefly noting the above key beneficial effect of INI in your Abstract.  However, it is up to you to decide if you wish to do so or not.) 

2.     “Currently, INI is widely used to treat Alzheimer's disease and mild cognitive impairment.” 

However, the Intranasal Insulin (INI) treatment is not an approved treatment for Alzheimer’s disease and mild cognitive impairment in most countries.  Therefore, the above sentence is incorrect as written.  Consequently, the quoted sentence must be changed to correctly state the current regulatory status of INI for Alzheimer’s and mild cognitive impairment.

3.     “One approaches to restoring the activity of the brain insulin system is the use of intranasally administered insulin (INI).”  The word “approaches” should be changed to “approach”.

The Introduction of this manuscript appropriately begins with this relevant and important statement:

“After the discovery of insulin by Frederick Banting and Charles Best in 1921, it was found that this hormone is produced by pancreatic β-cells and, when administered to the organism, has a hypoglycemic effect, controlling glucose homeostasis.”

4.     However, your manuscript is titled and focused on intranasal insulin as you note in the Introduction, “The role of the “first violin” here is assigned to intranasally administered insulin (INI), the use of which allows you to quickly increase the level of the hormone in the brain and, thereby, stimulate insulin signaling in neurons [16-19].” 

For this reason, you may wish to briefly state in the Introduction and reference the discovery and patenting by William H. Frey II of both direct intranasal delivery of therapeutics to the brain to treat neurological disorders and of intranasal insulin to treat Alzheimer’s, Parkinson’s, etc. 

Note that reference 16 by Crowe et al. (2018) of your manuscript referred to in the quote from the introduction above states the following in their published paper:

“William Frey II proposed and patented this method for direct delivery to the CNS for therapeutics following intranasal administration [10,11].” [[10] W.H. Frey II, Neurologic Agents for Nasal Administration to the Brain, World Intellectual Property Organization, 1991 1,991,007,947 A1, June 13.   [11] Frey WH II. Method of Administering Neurologic Agents to the Brain. US Patent 6, 180,603 B1, January 8, 1997.  [12] Frey WH II. Method for Administering Insulin to the Brain. US Patent 6,313,093 B1, November 6, 2001.]

Also, the following paper makes a similar statement:

Schiöth HB, Craft S, Brooks SJ, Frey WH 2nd, Benedict C. Brain insulin signaling and Alzheimer's disease: current evidence and future directions. Molecular Neurobiology. 2012 Aug;46(1):4-10. DOI: 10.1007/s12035-011-8229-6. PMID: 22205300; PMCID: PMC3443484.

“In 1989, Frey first proposed the noninvasive intranasal method for bypassing the BBB to target therapeutic proteins, growth factors, and hormones (including insulin) to the brain to treat neurodegenerative disorders such as Alzheimer's disease [23, 24] and later expanded on the specific use of intranasal insulin to target the brain to treat Alzheimer's disease and other CNS disorders [25, 26].”  References: 23. Frey WH II (1997) Method of administering neurologic agents to the brain. US Patent 5,624,898 filed 1989 and issued April 29, 1997

24. Frey WH II (1991) Neurologic agents for nasal administration to the brain. PCT International Patent WO91/07947 filed 1990 and issued June 13, 1991

25. Frey WH II (2001) Method for administering insulin to the brain. US Patent 6,313,093 B1 filed 1999 and issued November 6, 2001

26. Jogani V, Jinturkar K, Vyas T, Misra A. Recent patents review on intranasal administration for CNS drug delivery. Recent Pat Drug Deliv Formul. 2008;2(1):25–40. doi: 10.2174/187221108783331429. [Abstract] [CrossRef] [Google Scholar]

5.     The Introduction further states, “Currently, the intranasal route of drug administration is being actively studied and used in the clinic to deliver a variety of hormones, growth factors, pharmacological agents, as well as vaccines and stem cells [20, 21].”

This statement is incorrect as intranasal delivery is not being used in the clinic to deliver stem cells and some of the other things you mention here.  Further, your references [20, 21] are not reasonable references in regard to the intranasal delivery of therapeutic cells, including stem cells, to the brain since this was first discovered, patented and published by Lusine Danielyan M.D. and colleagues as follows: [Frey, Danielyan and Gleiter (2012). Methods, pharmaceutical compositions and articles of manufacture for administering therapeutic cells to the animal central nervous system. U.S. Patent 8283160 B2 filed 2009 and issued October 9, 2012.  Danielyan, L., et al., Intranasal delivery of cells to the brain. Eur J Cell Biol, 2009. 88(6): p. 315-24.  Danielyan, L., et al., Therapeutic efficacy of intranasally delivered mesenchymal stem cells in a rat model of Parkinson disease. Rejuvenation Res, 2011. 14(1): p. 3-16.  Danielyan, L. et al., Intranasal delivery of bone marrow-derived mesenchymal stem cells, macrophages, and microglia to the brain in mouse models of Alzheimer's and Parkinson's disease. Cell Transplant. 2014, 23 Suppl 1:S123-39.]

6.     Lines 66-80 of the Introduction present a misleading and incorrect picture of the mechanism and pharmacokinetics and pharmacodynamics of intranasal insulin delivery to the brain in animals and humans.  Here is a portion of what is stated,

“After nebulization in the nasal cavity, insulin enters the nasal mucosa, where it is transported to the brain by an intracellular route using receptor-mediated endocytosis, as well as via paracellular or transcellular pathways. All of these pathways allow insulin to spread along the olfactory and trigeminal nerves and rapidly distribute ventrally and dorsally. After 15–30 minutes, insulin accumulation was shown in the olfactory bulbs, striatum, substantia nigra, brainstem, cerebellum, and, to a lesser extent, in the hippocampus and cerebral cortex [17,18].”

First, the authors need to recognize that while intranasal peptides and proteins, like insulin and IGF-I, can transport to the brain via both intraneuronal and extracellular pathways, the intracellular mechanism takes a couple hours to reach even the olfactory bulb and days to weeks to reach other brain regions such as the hippocampus.  Almost all published studies about intranasal insulin are only interested in and only investigating the rapid extracellular delivery along the olfactory and trigeminal neural pathways to the brain.  It is this extracellular pathway that delivers insulin to the CSF of humans within 10 minutes and to important brain structures in animals in less than 30 minutes. [A paper discussing this in regard to intranasal IGF-I is Thorne RG, Pronk GJ, Padmanabhan V, Frey WH 2nd. Delivery of insulin-like growth factor-I to the rat brain and spinal cord along olfactory and trigeminal pathways following intranasal administration. Neuroscience. 2004;127(2):481-96. doi: 10.1016/j.neuroscience.2004.05.029. PMID: 15262337.]   Even reference 17 of this manuscript which is cited above in the quoted text of the Introduction states the following: “The rapid rate of transport, as observed in human (Born et al., 2002), monkeys (Thorne et al., 2008), as well as rats in our study, strongly favor the extracellular mechanism, since the speed of retrograde axonal transport is rather slow (Maday et al., 2014).”  Consequently, this portion of the Introduction must be revised and corrected.

7.     Lines 81-83 of the Introduction state the following: “To date, there is a large number of both experimental and clinical evidence that INI is highly effective in the treatment of AD, Parkinson's disease and mild cognitive impairment, as well as neuropathies [25-41].”

Lines 96-98 state: “The successful use of INI in the clinic to treat AD and other cognitive dysfunctions 96 opens up broad prospects for its use in the treatment of other pathologies, the etiology 97 and pathogenesis of which are associated with impaired insulin signaling in the brain.”

This is clearly not true or is at a minimum a gross exaggeration of the clinical use of INI and the effectiveness of INI. Recognize that intranasal insulin has not been approved by the FDA in the USA for use by physicians and their patients or by regulatory agencies in most other countries. Further, there is no INI nasal spray being marketed by the pharmaceutical industry to treat Alzheimer’s, Parkinson’s or other brain disorders.  Further, the magnitude of the memory improvement for Alzheimer’s does not indicate that INI is highly effective, even though it has repeatedly been shown to significantly improve memory in multiple clinical trials. Finally, I think there has only been one clinical trial published for INI treatment of Parkinson’s disease.  (This is due in part to the fact that the pharmaceutical industry has not really focused on developing and marketing INI to treat Alzheimer’s, Parkinson’s and other brain disorders.)

On lines 487 and 488 of the manuscript, it states, “It should be noted that back in the 2010s, intranasally administered IGF-1 began to be used to correct damage caused by cerebral ischemia-reperfusion [180, 187-193].”  This is incorrect since your references 187 and 188 were both published in 2001.  So, “the 2010s” needs to be replaced with “2001”.

Lines 601-615 and subsequent portions present results of the authors preclinical studies with INI treatment of ischemia which are very interesting to this reviewer. 

This review is quite long and covers a lot of scientific areas related to INI treatment. I am not an expert in many of these areas and can thus not critically review all of them.

Bases on my comments above, the authors need to carefully revise and correct their manuscript being sure that their comments are consistent with what is known about INI and what is in the papers they reference. 

Author Response

RESPONSE TO REVIEWERS

Hot spots for intranasal insulin application

New title: “Hot spots for the use of intranasal insulin: cerebral ischemia, brain injury, diabetes mellitus, endocrine disorders and postoperative delirium”

Alexander O. Shpakov, Inna I. Zorina, and Kira V. Derkach

COMMON SECTION OF RESPONSE TO REVIEWERS

We are very grateful to the Reviewers for a detailed analysis of our review article and for the comments made. We sincerely hope that the explanation, additions and changes made by us based on their comments and remarks have improved and expanded our review article.

In accordance with the requirements of the reviewers, we significantly revised the review article and made the following main changes and additions to it (in more detail, the changes and additions are presented below in the extended response to comments of the Reviewers 1 and 2).

  1. As recommended by the reviewers, the title of the review article has been changed. The new title: “Hot spots for the use of intranasal insulin: cerebral ischemia, brain injury, diabetes mellitus, endocrine disorders and postoperative delirium”.
  2. In accordance with the recommendations of the reviewers, significant changes were made to the abstract, which made it possible to better present the content of the manuscript in it. Along with the Abstract, the Introduction has been significantly revised, which made it possible to avoid a number of insufficiently substantiated statements.
  3. 3. In accordance with the requirements of the reviewers, the manuscript provides more detailed and more accurate information on the intranasal administration of substances, including insulin, for which the Subsection 2.3 (The intranasal route of insulin delivery) has been added.
  4. Some redundant information has been removed from Sections 3 (Intranasal Insulin and Cerebral ischemia) and 4 (Intranasal insulin and brain injury) to make these sections more balanced. In this regard, the necessary changes have been made to the List of References.

According to requirement of Editor and Reviewers, all changes and additions to the text are highlighted in yellow. Changed reference numbers in the text are colored blue.

In conclusion, we once again thank the Reviewers for their comments, which allowed us to add important information to the review article, clarify many aspects of the prospects for the practical use of intranasal insulin, and remove some redundant information.

With best regards,

The Authors

RESPONSE TO REVIEWER 1

MAJOR REMARKS

1.   “A decrease in the activity of the insulin signaling system of the brain, due to both central insulin resistance and insulin deficiency, leads to neurodegeneration and impaired regulation of appetite, metabolism, endocrine functions.  This is due to the neuroprotective properties of brain insulin and its leading role in the functioning of brain signaling network responsible for the regulation of the nervous, endocrine and other systems”.

What the Abstract does not explicitly briefly state is the following: glucose metabolism is the main source of brain cell energy, glucose hypometabolism occurs in the brain in numerous neurodegenerative diseases, including Alzheimer’s and Parkinson’s resulting in a loss of brain cell energy and that intranasal insulin plays a leading role in stimulating glucose uptake and metabolism which increases the production of brain cell energy in the form of ATP and Phosphocreatine.  [You should mention and reference somewhere in your manuscript the following paper: Jauch-Chara K, et al. Intranasal insulin suppresses food intake via enhancement of brain energy levels in humans. Diabetes. 2012 Sep;61(9):2261-8. doi: 10.2337/db12-0025. Epub 2012 May 14. PMID: 22586589; PMCID: PMC3425399.  (I suggest you consider briefly noting the above key beneficial effect of INI in your Abstract.  However, it is up to you to decide if you wish to do so or not.)

Response: We agree that the effect of brain insulin on glucose homeostasis in the CNS (uptake and metabolism of glucose, its involvement in energy processes in brain cells) is the most important component of the therapeutic effect of intranasal insulin on the nervous system and the body as a whole. This is discussed by us in the text, including in sufficient detail in Sections 3 (Intranasal insulin and brain ischemia) and 4 (Intranasal insulin and brain injury), when it comes to the restorative potential of INI in relation to disturbances in the energy status of brain cells and hypometabolic state caused by cerebral ischemia/reperfusion and traumatic brain injury. However, as noted by the reviewer in his commentary, this important mechanism of action of the INI is not mentioned in the Abstract. In accordance with the recommendations of the reviewer, we have expanded and modified the corresponding sentence in the Abstract:

«This is due to the neuroprotective properties of brain insulin, its leading role in maintaining glucose homeostasis in the brain, as well as in the regulation of the brain signaling network responsible for the functioning of the nervous, endocrine and other systems.»

We have also made a corresponding clarification to the text on page 3: «These data indicate a pronounced neuroprotective effect of INI, its ability to increase the viability of neurons and glial cells, and restore the functional interaction between brain signaling systems, providing integration between different brain structures and improving the central regulation of physiological functions. They also indicate the ability of INI to improve metabolic processes in the brain, including glucose uptake and metabolism, which prevents the hypometabolic states that are characteristic of brain damage and neurodegenerative disorders.».

Reference (see below) to the ability of insulin to improve energy metabolism in the brain and thereby reduce appetite is included in the text to the new Subsection 2.3.

  1. Currently, INI is widely used to treat Alzheimer's disease and mild cognitive impairment.” 

However, the Intranasal Insulin (INI) treatment is not an approved treatment for Alzheimer’s disease and mild cognitive impairment in most countries.  Therefore, the above sentence is incorrect as written.  Consequently, the quoted sentence must be changed to correctly state the current regulatory status of INI for Alzheimer’s and mild cognitive impairment.

Response: Thank you very much for your comment. Accordingly, we have modified our statement to indicate that INI is considered a promising drug for the treatment of Alzheimer's disease and mild cognitive impairment.

"INI is currently being considered as a promising drug to treat Alzheimer's disease and mild cognitive impairment."

  1. “One approaches to restoring the activity of the brain insulin system is the use of intranasally administered insulin (INI).”  The word “approaches” should be changed to “approach”.

The Introduction of this manuscript appropriately begins with this relevant and important statement:

“After the discovery of insulin by Frederick Banting and Charles Best in 1921, it was found that this hormone is produced by pancreatic β-cells and, when administered to the organism, has a hypoglycemic effect, controlling glucose homeostasis.”

Response: In the Abstract, a correction has been made to the sentence: “One of the approaches to restore the activity of the insulin system of the brain is the use of intranasally administered insulin (INI).” In this context, the word “approach” is used in the plural (“one of”).

As recommended by the reviewer, an important statement about the discovery of insulin and its functions by Banting and Best has been added to the preface to the manuscript:

“After the discovery of insulin by Frederick Banting and Charles Best in 1921, it was found that this hormone is produced by pancreatic β-cells and, when administered to the organism, has a hypoglycemic effect, controlling glucose homeostasis.”

  1. However, your manuscript is titled and focused on intranasal insulin as you note in the Introduction, “The role of the “first violin” here is assigned to intranasally administered insulin (INI), the use of which allows you to quickly increase the level of the hormone in the brain and, thereby, stimulate insulin signaling in neurons [16-19].” 

For this reason, you may wish to briefly state in the Introduction and reference the discovery and patenting by William H. Frey II of both direct intranasal delivery of therapeutics to the brain to treat neurological disorders and of intranasal insulin to treat Alzheimer’s, Parkinson’s, etc. 

Note that reference 16 by Crowe et al. (2018) of your manuscript referred to in the quote from the introduction above states the following in their published paper:

“William Frey II proposed and patented this method for direct delivery to the CNS for therapeutics following intranasal administration [10,11].” [[10] W.H. Frey II, Neurologic Agents for Nasal Administration to the Brain, World Intellectual Property Organization, 1991 1,991,007,947 A1, June 13.   [11] Frey WH II. Method of Administering Neurologic Agents to the Brain. US Patent 6, 180,603 B1, January 8, 1997.  [12] Frey WH II. Method for Administering Insulin to the Brain. US Patent 6,313,093 B1, November 6, 2001.]

Also, the following paper makes a similar statement:

Schiöth HB, Craft S, Brooks SJ, Frey WH 2nd, Benedict C. Brain insulin signaling and Alzheimer's disease: current evidence and future directions. Molecular Neurobiology. 2012 Aug;46(1):4-10. DOI: 10.1007/s12035-011-8229-6. PMID: 22205300; PMCID: PMC3443484.

“In 1989, Frey first proposed the noninvasive intranasal method for bypassing the BBB to target therapeutic proteins, growth factors, and hormones (including insulin) to the brain to treat neurodegenerative disorders such as Alzheimer's disease [23, 24] and later expanded on the specific use of intranasal insulin to target the brain to treat Alzheimer's disease and other CNS disorders [25, 26].”  References: 23. Frey WH II (1997) Method of administering neurologic agents to the brain. US Patent 5,624,898 filed 1989 and issued April 29, 1997

  1. Frey WH II (1991) Neurologic agents for nasal administration to the brain. PCT International Patent WO91/07947 filed 1990 and issued June 13, 1991
  2. Frey WH II (2001) Method for administering insulin to the brain. US Patent 6,313,093 B1 filed 1999 and issued November 6, 2001
  3. Jogani V, Jinturkar K, Vyas T, Misra A. Recent patents review on intranasal administration for CNS drug delivery. Recent Pat Drug Deliv Formul. 2008;2(1):25–40. doi: 10.2174/187221108783331429. [Abstract] [CrossRef] [Google Scholar]

Response: Thank you very much for your comments on Frey's pioneering work on intranasal insulin administration in more detail, which is important for understanding the development of a methodology for the use of INI for the treatment of neurodegenerative diseases and cognitive deficits. We have prepared and included in the manuscript a separate subsection (2.3) on the intranasal method of administering substances and, first of all, insulin, where all the recommended works are given and analyzed, including Frey's three fundamental patents on the intranasal method of drug delivery.

  1. Introduction further states, “Currently, the intranasal route of drug administration is being actively studied and used in the clinicto deliver a variety of hormones, growth factors, pharmacological agents, as well as vaccines and stem cells [20, 21].”

This statement is incorrect as intranasal delivery is not being used in the clinic to deliver stem cells and some of the other things you mention here.  Further, your references [20, 21] are not reasonable references in regard to the intranasal delivery of therapeutic cells, including stem cells, to the brain since this was first discovered, patented and published by Lusine Danielyan M.D. and colleagues as follows: [Frey, Danielyan and Gleiter (2012). Methods, pharmaceutical compositions and articles of manufacture for administering therapeutic cells to the animal central nervous system. U.S. Patent 8283160 B2 filed 2009 and issued October 9, 2012.  Danielyan, L., et al., Intranasal delivery of cells to the brain. Eur J Cell Biol, 2009. 88(6): p. 315-24.  Danielyan, L., et al., Therapeutic efficacy of intranasally delivered mesenchymal stem cells in a rat model of Parkinson disease. Rejuvenation Res, 2011. 14(1): p. 3-16.  Danielyan, L. et al., Intranasal delivery of bone marrow-derived mesenchymal stem cells, macrophages, and microglia to the brain in mouse models of Alzheimer's and Parkinson's disease. Cell Transplant. 2014, 23 Suppl 1:S123-39.]

Response: We agree with the reviewer that the intranasal route of administration should not be overestimated for some substances, such as stem cells and antibodies, and their consideration is redundant in the review concerning intranasal insulin. Previous cited references (20, 21) are more concerned with the general principles of the intranasal route of delivery of substances than with the description of specific classes of intranasally administered substances, and therefore have been replaced by other works. It is important that in the new version of the manuscript, the intranasal administration of drugs, primarily insulin, is considered in an additional subsection 2.3.

In accordance with the reviewer's comment, we removed "antibodies" and "stem cells" from sentence and softened our statement about the use of the intranasal route of administration of various substances in the clinic. The following text (the second and third paragraphs on page 2) has been partially moved to the additional Subsection 2.3 on intranasal administration, and the remaining sentences have been modified accordingly (see below).

 “Currently, the intranasal route of drug administration is being actively studied and considered for use in the clinic for the delivery of various hormones, growth factors, and pharmacological agents [20, 21]. The pharmacokinetics and pharmacodynamics of INI have been extensively studied in the rodent models [16-19] and in human [22]. INI almost does not enter the bloodstream and, as a result, does not cause a significant decrease in glucose levels at the periphery, which indicates that there is no risk of acute hypoglycemic episodes that often occur with subcutaneous and intravenous insulin injections and dangerous to diabetic patients [17, 19, 23, 24].”

  1. Patel, D.; Patel, B.; Wairkar, S. Intranasal delivery of biotechnology-based therapeutics. Drug Discov. Today 2022, 27, 103371. doi: 10.1016/j.drudis.2022.103371.
  2. Sharma, M.; Waghela, S.; Mhatre, R.; Saraogi, G.K. A Recent Update on Intranasal Delivery of High Molecular Weight Proteins, Peptides, and Hormones. Curr. Pharm. Des. 2021, 27, 4279-4299. doi: 10.2174/1381612827666210820100723

6. Lines 66-80 of the Introduction present a misleading and incorrect picture of the mechanism and pharmacokinetics and pharmacodynamics of intranasal insulin delivery to the brain in animals and humans.  Here is a portion of what is stated,

“After nebulization in the nasal cavity, insulin enters the nasal mucosa, where it is transported to the brain by an intracellular route using receptor-mediated endocytosis, as well as via paracellular or transcellular pathways. All of these pathways allow insulin to spread along the olfactory and trigeminal nerves and rapidly distribute ventrally and dorsally. After 15–30 minutes, insulin accumulation was shown in the olfactory bulbs, striatum, substantia nigra, brainstem, cerebellum, and, to a lesser extent, in the hippocampus and cerebral cortex [17,18].”

First, the authors need to recognize that while intranasal peptides and proteins, like insulin and IGF-I, can transport to the brain via both intraneuronal and extracellular pathways, the intracellular mechanism takes a couple hours to reach even the olfactory bulb and days to weeks to reach other brain regions such as the hippocampus.  Almost all published studies about intranasal insulin are only interested in and only investigating the rapid extracellular delivery along the olfactory and trigeminal neural pathways to the brain.  It is this extracellular pathway that delivers insulin to the CSF of humans within 10 minutes and to important brain structures in animals in less than 30 minutes. [A paper discussing this in regard to intranasal IGF-I is Thorne RG, Pronk GJ, Padmanabhan V, Frey WH 2nd. Delivery of insulin-like growth factor-I to the rat brain and spinal cord along olfactory and trigeminal pathways following intranasal administration. Neuroscience. 2004;127(2):481-96. doi: 10.1016/j.neuroscience.2004.05.029. PMID: 15262337.]   Even reference 17 of this manuscript which is cited above in the quoted text of the Introduction states the following: “The rapid rate of transport, as observed in human (Born et al., 2002), monkeys (Thorne et al., 2008), as well as rats in our study, strongly favor the extracellular mechanism, since the speed of retrograde axonal transport is rather slow (Maday et al., 2014).”  Consequently, this portion of the Introduction must be revised and corrected.

Response: We are grateful to the reviewer for his remarks about the differences in the rate of insulin delivery using different mechanisms, which we did not pay appropriate. In accordance with this, we have considered these issues in more detail in the additional subsection 2.3 on intranasal administration, and this text has been removed from the indicated block in the Introduction. See also our response to Comment 5 above.

  1. Lines 81-83 of the Introduction state the following: “To date, there is a large number of both experimental and clinical evidence that INI is highly effective in the treatment of AD, Parkinson's disease and mild cognitive impairment, as well as neuropathies [25-41].”

Lines 96-98 state: “The successful use of INI in the clinic to treat AD and other cognitive dysfunctions 96 opens up broad prospects for its use in the treatment of other pathologies, the etiology 97 and pathogenesis of which are associated with impaired insulin signaling in the brain.” 

This is clearly not true or is at a minimum a gross exaggeration of the clinical use of INI and the effectiveness of INI. Recognize that intranasal insulin has not been approved by the FDA in the USA for use by physicians and their patients or by regulatory agencies in most other countries. Further, there is no INI nasal spray being marketed by the pharmaceutical industry to treat Alzheimer’s, Parkinson’s or other brain disorders.  Further, the magnitude of the memory improvement for Alzheimer’s does not indicate that INI is highly effective, even though it has repeatedly been shown to significantly improve memory in multiple clinical trials. Finally, I think there has only been one clinical trial published for INI treatment of Parkinson’s disease.  (This is due in part to the fact that the pharmaceutical industry has not really focused on developing and marketing INI to treat Alzheimer’s, Parkinson’s and other brain disorders.)

Response: We agree that at present, despite the large amount of experimental data, there are relatively few clinical observations on the effectiveness of INI in Alzheimer's disease and other neurodegenerative diseases, although INI shows a pronounced restorative effect on cognitive deficits in these diseases. According to our data, in the US, intranasal insulin has the following FDA Status: Alzheimer's Disease (Phase 2/3), Mild Cognitive Impairment (Phase 2), Parkinson's Disease (Phase 2), and Multiple System Atrophy (Phase 2). In other countries, the situation is not much different.

We share the reviewer's point of view that this is largely due to the insufficient attention of pharmaceutical companies to the use of INI in medicine, which is sad, since it has a very powerful therapeutic potential. We very much hope that our research will make a certain contribution to the general intensive efforts of scientists working in the field of studying and using the INI, to actively implement it in clinical practice.

In accordance with the reviewer's comments, we have changed the emphasis in these sentences indicated by reviewer to avoid overestimating the clinical significance of INI:

Lines 81-83, Introduction

«To date, there is a large amount of experimental data, as well as clinical observations, indicating the effectiveness of INI in the treatment of AD, Parkinson's disease and mild cognitive impairment, as well as neuropathies [25-41]».

Lines 96-98, Introduction

«Data on the effectiveness of INI in the treatment of AD and other cognitive dysfunctions open up prospects for its use in the treatment of other pathologies, which are also, at least in part, due to functional changes in insulin signaling in the brain».

MINOR REMARKS

  1. On lines 487 and 488 of the manuscript, it states, “It should be noted that back in the 2010s, intranasally administered IGF-1 began to be used to correct damage caused by cerebral ischemia-reperfusion [180, 187-193].”  This is incorrect since your references 187 and 188 were both published in 2001.  So, “the 2010s” needs to be replaced with “2001”.

Response:  Thanks a lot for the clarification. In accordance with it, we changed the proposal regarding the timing of the studies: “It should be noted that intranasally administered IGF-1 has been used for the correction of damage caused by cerebral ischemia-reperfusion for 20 years [180, 187-193].” Thus, "2010s" was replaced with "for 20 years".

  1. Lines 601-615 and subsequent portions present results of the author’s preclinical studies with INI treatment of ischemia, which are very interesting to this reviewer. 

Response: We are very grateful to the reviewer for an important comment regarding our results on the use of intranasal insulin to prevent neurodegeneration during ischemia-reperfusion. We plan not only to continue, but also to expand these studies.

Information is also provided in the attached file.

Reviewer 2 Report

The manuscript entitled " Hot spots for intranasal insulin application" aimed to highlight the importance of insulin in the brain and its normal functions and then compiled the latest evidence regarding the bidirectional interplay between disrupted insulin signaling and pathophysiology of the brain disorders. in addition, the results of clinical and experimental studies regarding positive effects of intranasal insulin in cerebral ischemia and trauma brain injury, different types of DM, and disorders caused by anesthesia as well as HPG and HPT neuroendocrine axis. In spite of the fact that this is a comprehensive and timely review, I believe that it is somehow unfocused and unbalanced. As a starting point, the title and abstract do not accurately reflect the content of the MS. While authors stated that the focus of this work is ” INI in cerebral ischemia and trauma brain injury, different types of DM, and disorders caused by anesthesia” the title is written in a broader sense. In addition, the MS contain sections about the INI and HPG and HPT neuroendocrine axis, but nothing is mentioned in the abstract!!!

The second important point is that the MS is not balanced, there are some redundant sections and some important subjects are missing. for example, the authors wrote a lengthy description about the insulin signaling and pathways, while there is no section for introducing the intranasal administration of drugs, and its characteristics. For another example, detailed cellular mechanisms of cerebral ischemia and trauma brain injury, different types of DM, and disorders caused by anesthesia are written in unnecessary details, while the focus of study is promising effects of INI in these disorders. These parts could be found elsewhere and undermine the focus of the study.

There are some typo errors in MS, which should be revised, for instance, In line 151, a citation is written in two different style “…..insulin transcytosis [57] (Yu et al., 2006). Insulin transcytosis through….”

Author Response

RESPONSE TO REVIEWERS

Hot spots for intranasal insulin application

New title: “Hot spots for the use of intranasal insulin: cerebral ischemia, brain injury, diabetes mellitus, endocrine disorders and postoperative delirium”

Alexander O. Shpakov, Inna I. Zorina, and Kira V. Derkach

COMMON SECTION OF RESPONSE TO REVIEWERS

We are very grateful to the Reviewers for a detailed analysis of our review article and for the comments made. We sincerely hope that the explanation, additions and changes made by us based on their comments and remarks have improved and expanded our review article.

In accordance with the requirements of the reviewers, we significantly revised the review article and made the following main changes and additions to it (in more detail, the changes and additions are presented below in the extended response to comments of the Reviewers 1 and 2).

  1. As recommended by the reviewers, the title of the review article has been changed. The new title: “Hot spots for the use of intranasal insulin: cerebral ischemia, brain injury, diabetes mellitus, endocrine disorders and postoperative delirium”.
  2. In accordance with the recommendations of the reviewers, significant changes were made to the abstract, which made it possible to better present the content of the manuscript in it. Along with the Abstract, the Introduction has been significantly revised, which made it possible to avoid a number of insufficiently substantiated statements.
  3. 3. In accordance with the requirements of the reviewers, the manuscript provides more detailed and more accurate information on the intranasal administration of substances, including insulin, for which the Subsection 2.3 (The intranasal route of insulin delivery) has been added.
  4. Some redundant information has been removed from Sections 3 (Intranasal Insulin and Cerebral ischemia) and 4 (Intranasal insulin and brain injury) to make these sections more balanced. In this regard, the necessary changes have been made to the List of References.

According to requirement of Editor and Reviewers, all changes and additions to the text are highlighted in yellow. Changed reference numbers in the text are colored blue.

In conclusion, we once again thank the Reviewers for their comments, which allowed us to add important information to the review article, clarify many aspects of the prospects for the practical use of intranasal insulin, and remove some redundant information.

With best regards,

The Authors

RESPONSE TO REVIEWER 2

MAJOR REMARKS

  1. In spite of the fact that this is a comprehensive and timely review, I believe that it is somehow unfocused and unbalanced. As a starting point, the title and abstract do not accurately reflect the content of the MS. While authors stated that the focus of this work is ”INI in cerebral ischemia and trauma brain injury, different types of DM, and disorders caused by anesthesia” the title is written in a broader sense. In addition, the MS contain sections about the INI and HPG and HPT neuroendocrine axis, but nothing is mentioned in the abstract!!!

Response: We agree that the title and abstract could be improved to better match the content of the review.

We have changed the title as follows, considering that in this case it contains all the necessary information regarding the content of the review.

The old title of the review «Hot spots for intranasal insulin application» was replaced by «Hot spots for the use of intranasal insulin: cerebral ischemia, brain injury, diabetes mellitus, endocrine disorders and postoperative delirium».

In the Abstract, in accordance with the reviewer's comment, information was added on the effect of INI on the functional activity of the gonadal and thyroid axes, which were the subject of our close study, but, due to the limitations of the abstract, were not presented in it.

«At the same time, much attention has recently been paid to the prospects of using INI for the treatment of cerebral ischemia, traumatic brain injuries, postoperative delirium (after anesthesia), as well as diabetes mellitus and its complications, including dysfunctions in the gonadal and thyroid axes.»

  1. The second important point is that the MS is not balanced, there are some redundant sections and some important subjects are missing. For example, the authors wrote a lengthy description about the insulin signaling and pathways, while there is no section for introducing the intranasal administration of drugs, and its characteristics.

Response: We agree with the reviewer that a relatively short description of the intranasal route of administration, including for insulin, in the Introduction (two paragraphs) was not enough, especially since the review is devoted specifically to INI. Based on this observation, we have prepared an additional subsection on intranasal administration of insulin to the Section 2 (Subsection 2.3, The intranasal route of insulin delivery). Particular attention is paid to the pioneering work of William Frey II, who more than 30 years ago first proposed and studied the intranasal method of delivering various biologically active substances to the brain, including insulin, which can then be used to treat neurodegenerative diseases, including Alzheimer's disease.

The new subsection 2.3 is entirely yellow.

In the course of its preparation, new references were added (also highlighted in yellow), which were taken into account when renumbering references throughout the manuscript.

  1. For another example, detailed cellular mechanisms of cerebral ischemia and trauma brain injury, different types of DM, and disorders caused by anesthesia are written in unnecessary details, while the focus of study is promising effects of INI in these disorders. These parts could be found elsewhere and undermine the focus of the study.

Response: We agree that there is some redundant information in the Sections 3 (Intranasal insulin and brain ischemia) and 4 (Intranasal insulin and brain injury). It seemed to us that this information could help in understanding many aspects and prospects for the use of INI in these pathologies, and could also emphasize the need to develop new approaches to their treatment, including those based on INI. We have carefully analyzed these sections and deleted some fragments that add little to the analysis and evaluation of the available data on the use of INI in cerebral ischemia and brain injury.

The positions where text reductions were made are highlighted in yellow (pages 9, 16 and 22). 

Since in the process of shortening the text some references were lost, this was taken into account when changing the numbering of references throughout the manuscript.

MINOR REMARKS

  1. There are some typo errors in MS, which should be revised, for instance, In line 151, a citation is written in two different style “…..insulin transcytosis [57] (Yu et al., 2006). Insulin transcytosis through….”

Response: We are grateful to the reviewer for the remark regarding the two different styles of use of the term insulin transcytosis. Required changes have been made because the phrase «Insulin transcytosis through…» is not correct.

Corrected version:

«Insulin transcytosis is regulated by hormones, lipids, and vasodilation factors.»

Information is also provided in the attached file.

Round 2

Reviewer 1 Report

The authors have made many changes that have greatly improved their manuscript.  The following additional changes need to be made:

1.     Revise on page 2:  “INI almost does not enter the bloodstream and, as a result, does not cause a significant decrease in glucose levels at the periphery, which indicates that there is no risk of acute hypoglycemic episodes that often occur with subcutaneous and intravenous insulin injections and dangerous to diabetic patients [17, 19, 23, 24].”

to read as follows:

“INI, at doses used in human clinical trials to treat patients with Type 2 Diabetes, MCI or Alzheimer’s disease, does not significantly alter the blood levels of insulin or glucose, which indicates that there is little risk of acute hypoglycemic episodes that can occur with subcutaneous and intravenous insulin injections that may be dangerous.”

and cite references to appropriate articles, not those used above that are not appropriate.  The following references from the current version of your manuscript could be cited here [25, 27, 28, 33, 34].

2.     Revise on page 9:  “Significant progress in the study of the mechanisms of penetration of INI through the 400 BBB, its pharmacokinetics and pharmacodynamics, as well as the targets of INI in the 401 brain has been achieved through the use of animal models [16-19, 134].”           

to read as follows:

“Significant progress in the study of the INI mechanisms involved in bypassing the BBB, its pharmacokinetics and pharmacodynamics, as well as the targets of INI in the brain has been achieved through the use of animal models [16-19, 134].”

3.      Revise on page 11:  “It should be noted that back in the 2010s, intranasally administered IGF-1 began to be used to correct damage caused by cerebral ischemia-reperfusion [186, 193-199].”      (Note: this error was noted in my previous review and was not corrected even though the authors apparently intended to correct it.  References 193, 194 and 195 were not in the 2010s, but rather in 2001 and 2004.  The authors need to correct this 2010s wording.)

Author Response

THE SECOND RESPONSE TO REVIEWERS

Hot spots for intranasal insulin application

New title: “Hot spots for the use of intranasal insulin: cerebral ischemia, brain injury, diabetes mellitus, endocrine disorders and postoperative delirium”

Alexander O. Shpakov, Inna I. Zorina, and Kira V. Derkach

COMMON SECTION OF RESPONSE TO REVIEWERS

We are very grateful to the Reviewers for a repeated detailed analysis of our review article and for appreciating the changes and corrections we made after the first round of peer review. We sincerely hope that the changes made by us based on their comments after the second round of peer review have improved our review article.

The changes are presented below in the response to comments of the Reviewer 1.

According to requirement of Editor and Reviewers, all changes and additions to the text are highlighted in green.  

Additionally, in the Section “Funding” (Page 29) we have removed some of the support information. Deleted text is «…(the sections 2, 3 and 5) and by the Sechenov Institute of Evolutionary Physiology and Biochemistry of the Russian Academy of Sciences research program No. 075-0152-22-00 (the sections 4, 6 and 7)». The finalized version of the section “Funding” is highlighted in green.

With best regards,

The Authors

Response to Reviewer 1 (the second round of peer review)

Minor comments

  1. Revise on page 2: “INI almost does not enter the bloodstream and, as a result, does not cause a significant decrease in glucose levels at the periphery, which indicates that there is no risk of acute hypoglycemic episodes that often occur with subcutaneous and intravenous insulin injections and dangerous to diabetic patients [17, 19, 23, 24].”

to read as follows:

“INI, at doses used in human clinical trials to treat patients with Type 2 Diabetes, MCI or Alzheimer’s disease, does not significantly alter the blood levels of insulin or glucose, which indicates that there is little risk of acute hypoglycemic episodes that can occur with subcutaneous and intravenous insulin injections that may be dangerous.”

and cite references to appropriate articles, not those used above that are not appropriate.  The following references from the current version of your manuscript could be cited here [25, 27, 28, 33, 34].

Response: Thank you very much for your comment, the necessary changes have been made.

  1. Revise on page 9: “Significant progress in the study of the mechanisms of penetration of INI through the BBB, its pharmacokinetics and pharmacodynamics, as well as the targets of INI in the brain has been achieved through the use of animal models [16-19, 134].”           

to read as follows:

“Significant progress in the study of the INI mechanisms involved in bypassing the BBB, its pharmacokinetics and pharmacodynamics, as well as the targets of INI in the brain has been achieved through the use of animal models [16-19, 134].”

Response: Thank you very much for your comment, the necessary changes have been made.

  1. Revise on page 11: “It should be noted that back in the 2010s, intranasally administered IGF-1 began to be used to correct damage caused by cerebral ischemia-reperfusion [186, 193-199].”   (Note: this error was noted in my previous review and was not corrected even though the authors apparently intended to correct it.  References 193, 194 and 195 were not in the 2010s, but rather in 2001 and 2004.  The authors need to correct this 2010swording.)

Response: We apologize that we did not make the text change given in the Response to the reviewer in the corrected text of the article. We changed the proposal regarding the timing of the studies: “It should be noted that intranasally administered IGF-1 has been used for the correction of damage caused by cerebral ischemia-reperfusion for 20 years [180, 187-193].” Thus, "2010s" was replaced with "for 20 years".

Reviewer 2 Report

The authors have addressed my concerns, and I recommend publication in the journal

Author Response

THE SECOND RESPONSE TO REVIEWERS

Hot spots for intranasal insulin application

New title: “Hot spots for the use of intranasal insulin: cerebral ischemia, brain injury, diabetes mellitus, endocrine disorders and postoperative delirium”

Alexander O. Shpakov, Inna I. Zorina, and Kira V. Derkach

COMMON SECTION OF RESPONSE TO REVIEWERS

We are very grateful to the Reviewers for a repeated detailed analysis of our review article and for appreciating the changes and corrections we made after the first round of peer review. We sincerely hope that the changes made by us based on their comments after the second round of peer review have improved our review article.

The changes are presented below in the response to comments of the Reviewer 1.

According to requirement of Editor and Reviewers, all changes and additions to the text are highlighted in green.  

With best regards,

The Authors